# Synergistic cancer immunotherapy combines MVA-CD40L induced innate and adaptive immunity with tumor targeting antibodies

José Medina-Echeverz[1,3], Maria Hinterberger[1,3], Marco Testori[1], Marlene Geiger[1], Raphael Giessel[1], Barbara Bathke[1], Ronny Kassub[1], Fabienne Gräbnitz[1], Giovanna Fiore[1], Sonia T. Wennier[1], Paul Chaplin[1], Mark Suter[2], Hubertus Hochrein [1,4]* & Henning Lauterbach [1,4]

Virus-based vaccines and appropriate costimulation potently enhance antigen-specific T cell immunity against cancer. Here we report the use of recombinant modified vaccinia virus Ankara (rMVA) encoding costimulatory CD40L against solid tumors. Therapeutic treatment with rMVA-CD40L-expressing tumor-associated antigens results in the control of established tumors. The expansion of tumor-specific cytotoxic CD8$^+$ T cells is essential for the therapeutic antitumor effects. Strikingly, rMVA-CD40L also induces strong natural killer (NK) cell activation and expansion. Moreover, the combination of rMVA-CD40L and tumor-targeting antibodies results in increased therapeutic antitumor efficacy relying on the presence of Fc receptor and NK cells. We describe a translationally relevant therapeutic synergy between systemic viral vaccination and CD40L costimulation. We show strengthened antitumor immune responses when both rMVA-CD40L-induced innate and adaptive immune mechanisms are exploited by combination with tumor-targeting antibodies. This immunotherapeutic approach could translate into clinical cancer therapies where tumor-targeting antibodies are employed.

[1] Bavarian Nordic GmbH, Fraunhoferstrasse 13, 82152 Planegg, Germany. [2] Vetsuisse Fakultät, Dekanat, Bereich Immunologie, Universität Zürich, Zürich, Switzerland. [3] These authors share co-authorship: José Medina-Echeverz, Maria Hinterberger. [4] These authors jointly supervised this work: Hubertus Hochrein, Henning Lauterbach. *email: hho@bavarian-nordic.com

T cell-based cancer immunotherapies, such as checkpoint inhibition or adoptive cell therapy dramatically changed the way of cancer treatment. Checkpoint-blocking antibodies such as anti-PD-1 or anti-CTLA-4 reinvigorate pre-existing T-cell responses against tumor-associated antigens (TAAs). Adoptive cell therapy works by expanding ex vivo a patient's own tumor-specific T cells or by genetically modifying T cells to express a chimeric antigen receptor (CAR) before expansion and reinfusion into the patient. Similar to CAR T cells, monoclonal antibodies (mAbs) such as Herceptin® or Rituximab® bind to surface molecules on cancer cells and target these cells for destruction. Another strategy under investigation are therapeutic cancer vaccines, which are designed to generate de novo and to expand TAA-specific CD4$^+$ and CD8$^+$ T cell responses. The success of vaccination is determined by the vaccine components chosen to increase the magnitude, breadth, and functionality of the induced immune responses. Multiple factors such as the tumor antigens encoded, targeting scaffold, use of adjuvants, tropism, or route of administration influence the therapeutic outcome of a vaccine[1]. Live viral vectors represent an excellent therapeutic vaccination moiety to deliver tumor antigens. Viral infection and death of infected cells trigger the release of danger- and pathogen-associated molecular patterns that activate antiviral defense mechanisms that tune up antitumor innate and adaptive immune responses depending on the nature of the viral scaffold[2].

Replication-competent viruses are currently being evaluated in multiple clinical trials, either by local or systemic delivery with the aim to target multiple tumors in one patient[3,4]. Despite their engineered tissue tropism, viral pharmacodynamics, pharmacokinetics, and possible pre-existing immunity, replication-competent viruses in cancer patients still raise safety concerns. Modified vaccinia virus Ankara (MVA) is an attenuated cytophatic strain that has been developed into a third-generation smallpox vaccine (MVA-BN, JYNNEOS, IMVAMUNE, and IMVANEX). MVA's replication deficiency in human cells and its clinically tested immunogenicity and safety in large numbers of clinical trials underline MVA as a safe prime candidate for therapeutic vaccine design. MVA-BN®-based vectors are currently being investigated as therapeutic cancer vaccines in various indications[5–7].

The TNF receptor family member CD40 is considered a costimulatory master switch of dendritic cells (DCs). CD40-CD40L-based interaction of DCs with activated CD4$^+$ T cells "license" DCs to prime CD8$^+$ T cell responses. This potent axis of immunity has been successfully evaluated in preclinical cancer settings[8]. After an initial wave of excitement, targeting the CD40-CD40L axis in clinical trials by the use of agonistic CD40 mAbs has been associated with immune-related adverse events[9]. We previously reported that MVA-BN® engineered to express CD40L efficiently activated DCs in vitro and in vivo, leading to enhanced effector and memory CD8$^+$ T cell and various innate immune responses[10]. Due to its vigorous immunogenicity and inhability to replicate, rMVA-CD40L represents a safe and promising anticancer vaccine.

In this study, we evaluate the therapeutic activity of rMVA-CD40L against established tumors. Single systemic immunization with CD40L-adjuvanted rMVA exerts strong objective therapeutic responses in various unrelated tumor models. This antitumor effect is based on the generation of non-exhausted, systemic antitumor CD8$^+$ T cells. Immunization with rMVA-CD40L modulates the production of proinflammatory cytokines, and expands and activates natural killer (NK) cells systemically. Therapeutic combination of rMVA-CD40L with antibodies targeting TAAs significantly improves antitumor responses against established, aggressive carcinomas relying on NK cell and Fc receptor-dependent mechanisms.

## Results

**Superior antitumor control upon rMVA-CD40L immunization.** Systemic immunization with poxviruses has been shown to effectively induce antitumor responses in various tumor models[11], which were further enhanced when adjuvanted with interleukin (IL)2 or IL12[12,13]. As CD40L expression by replication-deficient rMVA markedly enhanced systemic rMVA immunostimulatory capacities leading to improved protection against a lethal *Ectromelia* virus (ECTV) challenge[10], we wanted to assess the therapeutic effect of single intravenous administration of rMVA encoding CD40L against established tumors (Fig. 1a). A single immunization with an MVA vector encoding ovalbumin (OVA; referred to as rMVA) significantly induced tumor growth control in OVA-expressing B16 melanoma (Fig. 1b) and EG7.OVA lymphoma (Supplementary Fig. 2A) compared with phosphate-buffered saline (PBS)-treated mice. Interestingly, administration of MVA-OVA-CD40L (referred to as rMVA-CD40L) resulted in prolonged mouse survival in melanoma (Fig. 1c) and lymphoma, where 30% of the animals rejected their tumors (Supplementary Fig. 2B). In addition, a strong expansion of OVA$_{257-264}$-specific CD8$^+$ T cells was observed in the peripheral blood of tumor-bearing mice 7 days after immunization with rMVA vectors in both tumor models (Supplementary Fig. 2,C, D; see Supplementary Fig. 1 for flow cytometry gating strategies). Repeated administration of rMVA-CD40L did not increase antitumor responses against B16.OVA melanoma tumors (Supplementary Fig. 3).

Similar results were observed in MC38.WT, CT26.WT, and CT26.HER2 tumor-bearing mice after immunization with a CD40L-adjuvanted MVA encoding H-2K$^b$- and H-2L$^d$-restricted endogenous TAAs (MVA-TAA-CD40L) (Fig. 1d, f, respectively) or human epidermal growth factor 2 (HER2) (MVA-HER2-CD40L, Fig. 1h) respectively. Importantly, a significant increase in mouse survival was observed for both MVA-TAA-CD40L- and MVA-HER2-CD40L-treated groups in the beforementioned tumor models (Fig. 1e, g and i, respectively). Indeed, MVA-HER2-CD40L induced CT26.HER2 tumor regression in 100% of treated mice. Altogether, intravenous administration of rMVA-CD40L potentiates MVA-mediated antitumor immune responses in different mouse strains in various unrelated tumor models and using different model antigens.

**rMVA-CD40L induces antigen-specific CD8$^+$ T cells in tumors.** As CD8$^+$ T cell expansion in peripheral organs has been described as a main feature of systemic MVA immunizations[10,14,15], we sought to determine whether CD8$^+$ T cells could be found in the spleen and tumors 7 days after systemic immunization. Interestingly, rMVA-CD40L immunization resulted in significantly increased percentage of total CD8$^+$ T cells and specifically of activated OVA$_{257-264}$-specific CD8$^+$ T cells in the spleen and tumor (Fig. 2a, b). Of note, although rMVA treatment increased total CD8$^+$ T cells among leukocytes, only CD40L-adjuvanted rMVA selectively expanded antigen-specific CD8$^+$ T cells in the tumor microenvironment.

Lag3 and PD-1 are co-expressed on functionally impaired ("exhausted") tumor-infiltrating lymphocytes (TILs), thereby contributing to tumor-mediated immune suppression both in preclinical models[16,17] and in cancer patients[18]. As shown in Fig. 2c, expression of Lag3 and PD-1 by OVA$_{257-264}$-specific CD8$^+$ TILs is decreased when mice received rMVA and, more strikingly, when mice were immunized with rMVA-CD40L (Fig. 2c, d). The latter condition also demonstrated the most drastic reduction in T cells expressing high amounts of PD-1 and Lag3 simultaneously. This effect was also observed in

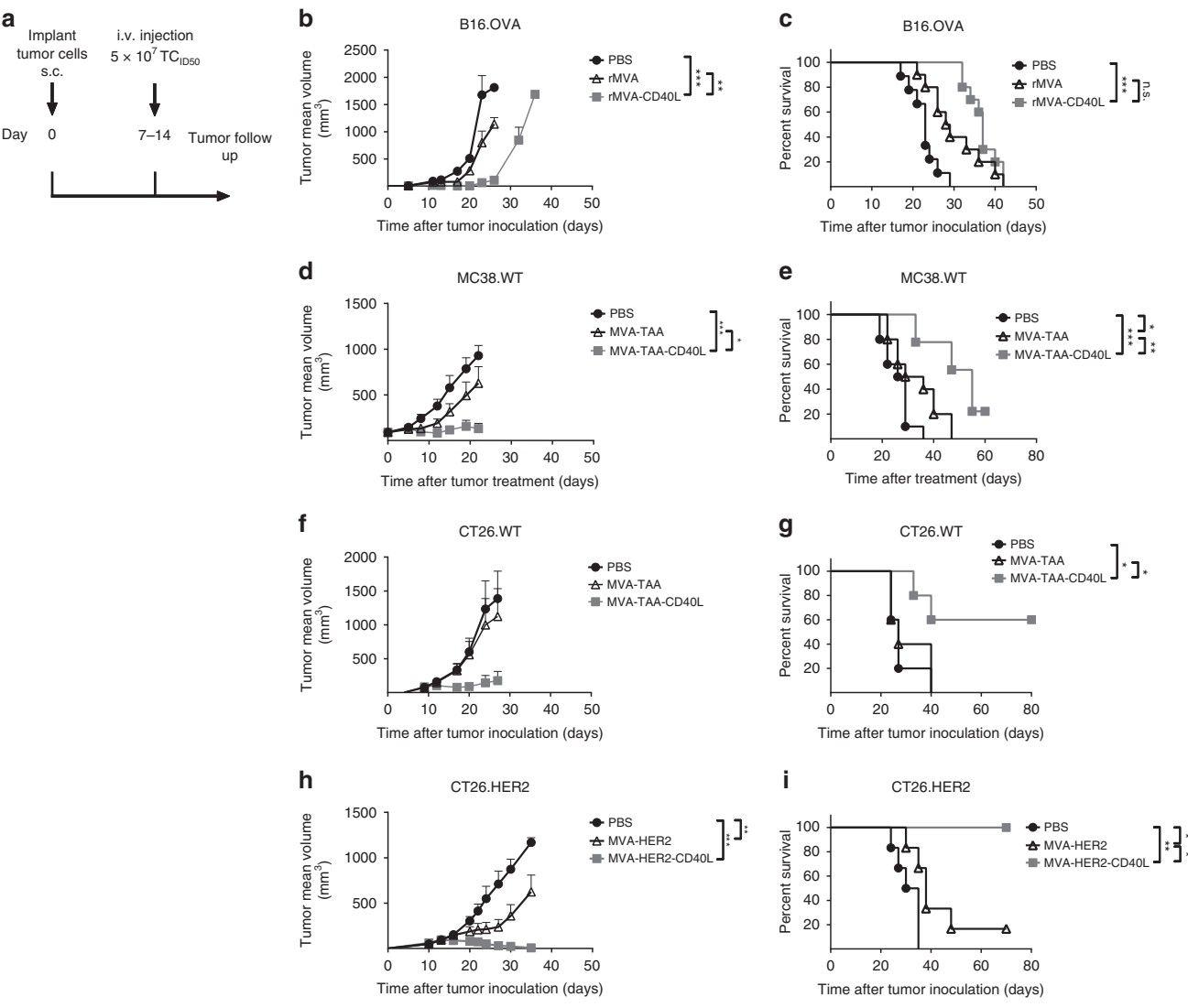

**Fig. 1** Therapeutic efficacy of rMVA-CD40L in unrelated, large, established tumor models. **a** Experimental layout: briefly, C57BL/6 (**b–e**) or Balb/c mice (**f–i**) received either B16.OVA (**b**, **c**), MC38.WT (**d**, **e**), CT26.WT (**f**, **g**) or CT26.HER2 (**h**, **i**) cells subcutaneously in the flank. Seven to 14 days later, when tumors were above 60 $mm^3$, mice were immunized intravenously either with PBS or with $5 \times 10^7$ $TCID_{50}$ of the mentioned rMVA viruses. **b**, **c** B16.OVA; **b** tumor size follow-up ($n = 5$ mice/group) and **c** overall survival ($n = 10$ mice/group) of mice injected either with PBS, MVA-OVA, or MVA-OVA-CD40L; **d**, **e** MC38.WT tumor-bearing mice were grouped 18 days after tumor cell inoculation, when tumors were above 90 $mm^3$; **d** tumor size follow-up ($n = 10$ mice/group) and **e** overall survival ($n = 10$ mice/group) of mice injected either with PBS, MVA-TAA, or MVA-TAA-CD40L until day 60 after treatment; **f**, **g** CT26.WT; **f** tumor size follow-up ($n = 5$ mice/group), and **g** overall survival ($n = 5$ mice/group) of mice injected either with PBS, MVA-TAA, or MVA-TAA-CD40L; **h**, **i** CT26.HER2; **h** tumor size follow-up ($n = 6$ mice/group), and **i** overall survival ($n = 6$ mice/group) of mice injected either with PBS or MVA-HER2-CD40L. **b**, **d**, **f**, and **h** Data are expressed as mean ± SEM. Panel **b** is representative of at least two independent experiments. Panel **c** represents overall survival of two merged independent experiments. The antitumor efficacy of MVA-HER2-CD40L in **h** and **i** has been tested in the CT26.HER2 tumor model in at least two independent experiments. One-way ANOVA at day 20 after tumor inoculation was performed on **b**, **d**, **f**, and **h**. Log-rank test on mouse survival was performed for **c**, **e**, **g**, and **i**. *$p < 0.05$; **$p < 0.01$; ***$p < 0.005$

tumor-infiltrating CD4+ T cells (Supplementary Fig. 4A), indicating that systemic rMVA immunization affects both tumor-infiltrating T cell subsets. Peripheral T cell responses in melanoma patients responding to PD-1 immune checkpoint blockade have been associated with increased expression of the proliferation marker Ki67[19,20]. Assuming a similar relationship between these two markers in TILs from mice immunized with rMVA and rMVA-CD40L, the co-expression of Ki67 and PD-1 of these cells was assessed. The proportion of Ki67+ in PD-1neg CD8+ T cells was higher in rMVA-CD40L-immunized tumor-bearing mice (Fig. 2e, f) compared with rMVA immunization and PBS (Fig. 2e, f). Tumor-infiltrating CD4+ T cells showed a

similar phenotype (Supplementary Fig. 4B,C), indicating that rMVA-CD40L immunization promotes TIL expansion and reinvigoration.

Regulatory T cells ($T_{reg}$) are important in tumor-induced immune suppression due to their deleterious effect in abrogating antitumor T cell responses. rMVA or rMVA-CD40L immunization results in decreased tumor-infiltrating $T_{reg}$ frequencies (Fig. 2g). As PD-1 is highly expressed on $T_{reg}$ and is pivotal for their function[21,22], PD-1 expression on tumor-infiltrating $T_{reg}$ was also assessed. Both rMVA or rMVA-CD40L immunization significantly decreased PD-1 expression on tumor-infiltrating $T_{reg}$ (Fig. 2h). Consistently, $OVA_{257-264}$-specific CD8+ T cell ($T_{eff}$) to

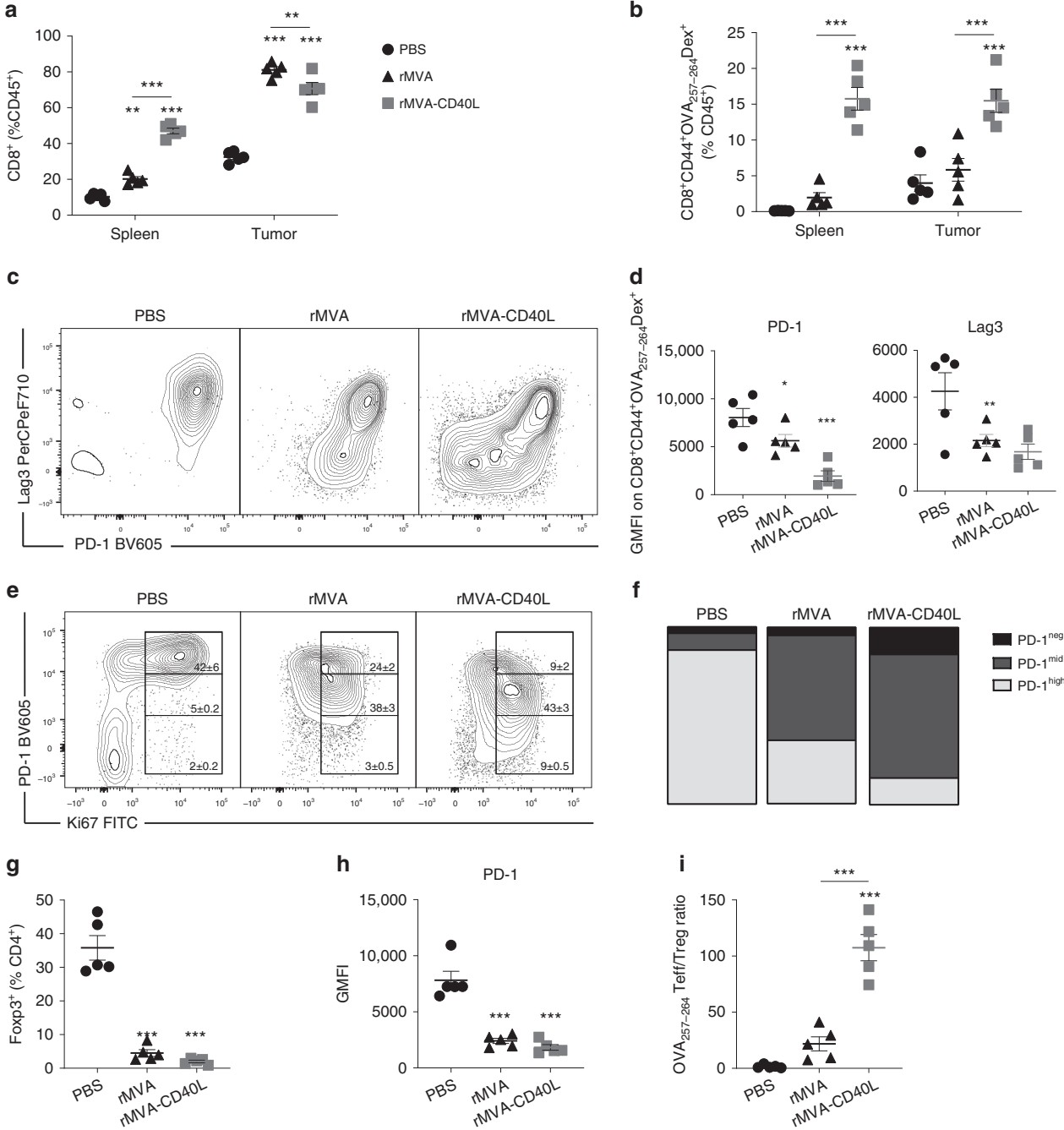

**Fig. 2** rMVA-CD40L increases intratumor T cell infiltration of non-exhausted CD8+ T cells. B16.OVA tumor bearers were immunized when tumors reached at least 50 mm³ in volume. Seven days later, mice were killed for further analysis ($n = 5$ mice/group). **a** Frequency of CD8+ T cells among leukocytes in the spleen and tumor tissues. **b** Distribution of OVA$_{257-264}$-specific CD8+ T cells in different organs upon immunization. **c** Representative dot plot of PD-1 and Lag3 co-expression in tumor-infiltrating OVA$_{257-264}$-specific CD8+ T cells. **d** GMFI of PD-1 and Lag3 on tumor-infiltrating OVA$_{257-264}$-specific CD8+ T cells. **e** Representative dot plot of Ki67 and PD-1 expression on tumor-infiltrating CD8+ T cells showing mean ± SEM, representative of at least two independent experiments. **f** Bar chart representing PD-1 expression on Ki67+ TILs shown in **e**. **g** Frequency of tumor-infiltrating FoxP3+ T$_{reg}$ among CD4+ T cells. **h** GMFI of PD-1 on tumor-infiltrating T$_{reg}$. Data expressed as mean ± SEM, representative of at least two independent experiments. **i** OVA$_{257-264}$-specific CD8+ T cell (OVA$_{257-264}$ T$_{eff}$) to T$_{reg}$ ratio. Data expressed as mean ± SEM, representative of at least two independent experiments in **d**, **f**, **g**, **h**, and **i**. One-way ANOVA comparing treatment groups. **a**, **b** Two-way ANOVA comparing cell frequencies in analyzed organs upon treatment. Statistics were run comparing treatment vs. PBS. *$p < 0.05$; **$p < 0.01$; ***$p < 0.005$.

T$_{reg}$ ratio was significantly increased in rMVA-CD40L immunization compared with the other treatments (Fig. 2i).

**Antitumor effect of rMVA-CD40L depends on CD8+ T cells.** We next addressed the contribution of CD8+ T cells in the antitumor effect following rMVA-CD40L immunization. Antibody depletion of CD8+ T cells resulted in complete loss of tumor growth arrest by either rMVA or rMVA-CD40L immunization (Fig. 3a, b), pointing to an essential role of CD8+ T cells in controlling tumor growth.

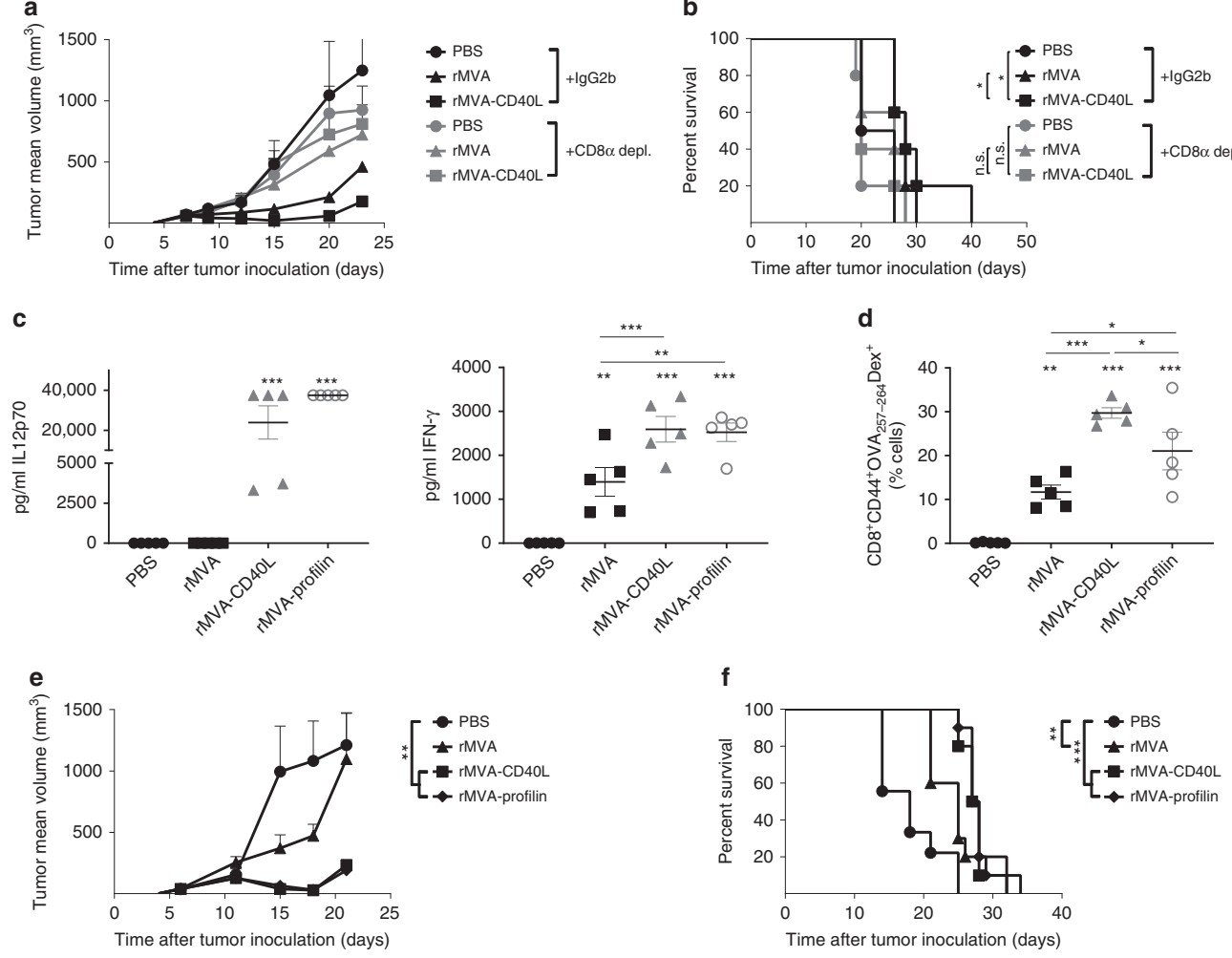

**Fig. 3** Role of CD8[+] T cells during rMVA-CD40L-induced tumor growth control. **a**, **b** CD8 depletion in B16.OVA tumor-bearing mice. When B16.OVA tumors were above 50 mm[3], mice received PBS or were immunized with $5 \times 10^7$ TCID[50] of either rMVA or rMVA-CD40L. Where indicated, mice received at days −2, 2, 6, and 10 after immunization 200 µg of either IgG2b or anti-CD8 antibody i.p. **a** Tumor size follow-up ($n = 5$ mice/group) and **b** overall survival. **a** Representative growth of PBS and rMVA-CD40L-treated groups in at least two independent experiments. **b** Represents overall survival of one independent experiment. Data in **a** expressed as mean ± SEM. **c–f** C57BL/6 mice bearing 50 mm[3] B16.OVA tumors ($n = 5$ mice/group) received PBS or were immunized with $5 \times 10^7$ TCID[50] of either rMVA, rMVA-CD40L, or rMVA-Profilin. **c** Quantitative expression of IL12p70 and IFN-γ in sera 6 h after systemic immunization. **d** Percentage of CD44[+]OVA[257-264]-specific CD8[+] T cells among peripheral blood leukocytes (PBL) 7 days after immunization. **e** Tumor size follow-up. **f** Overall survival. **c–e** Data representative of two independent experiments. **f** Survival of two merged independent experiments. Data in **a**, **c**, **d**, and **e** are expressed as mean ± SEM. One-way ANOVA was performed on figures **a**, **c**, **d**, and **e**. Log-rank test on mouse survival was performed for **b** and **f**. NS, nonsignificant; *$p < 0.05$; **$p < 0.01$; ***$p < 0.005$

Then, we sought to determine how this potent CD8[+] T cell response is induced. The generation of CD8[+] T cells against MVA-encoded antigens seems to rely on CD8[+] T cell cross-priming[23]. Activation of CD8α[+] cDCs results in large amounts of IL12p70, which thereby stimulates interferon-γ (IFN-γ) secretion. In addition to rMVA and rMVA-CD40L, rMVA expressing profilin (rMVA-Profilin)—a protein derived from the protozoan parasite *Toxoplasma gondii* that is specifically recognized by mouse CD8α[+] cDCs via TLR11 and TLR12[24–26]—was used to immunize tumor-bearing littermates. rMVA-CD40L and rMVA-Profilin immunization resulted in IL12p70 production and increased levels of IFN-γ in mice sera compared with rMVA (Fig. 3c). Similar to rMVA-CD40L, significantly higher expansion of OVA[257-264]-specific CD8[+] T cells in the peripheral blood 7 days after rMVA-Profilin compared with rMVA was observed (Fig. 3d). In addition, systemic immunization of B16.OVA tumor-bearing mice with rMVA-Profilin controlled tumor

growth and prolonged mouse survival comparable to that effect of systemic rMVA-CD40L (Fig. 3e, f).

**rMVA-CD40L enhances systemic NK cell activation.** NK cells play an important role in the host defense against viral infections[27]. Indeed, intravenous rMVA immunization induces the secretion of cytokines such as IL18 and IFN-α[10], key for NK cell expansion, activation, and homeostasis[28,29]. We hypothesized that intravenous rMVA immunization might result in systemic priming of NK cells. We thus determined the frequency of NK cells in different organs at days 1 and 4 after immunization (Fig. 4a). The frequency of NK cells in the spleen 1 day after immunization was significantly decreased, whereas a large increase was observed in the liver and in the lung. Interestingly, the expression of Ki67 remained unaltered during this time point among spleen-, liver-, and lung-infiltrating NK cells

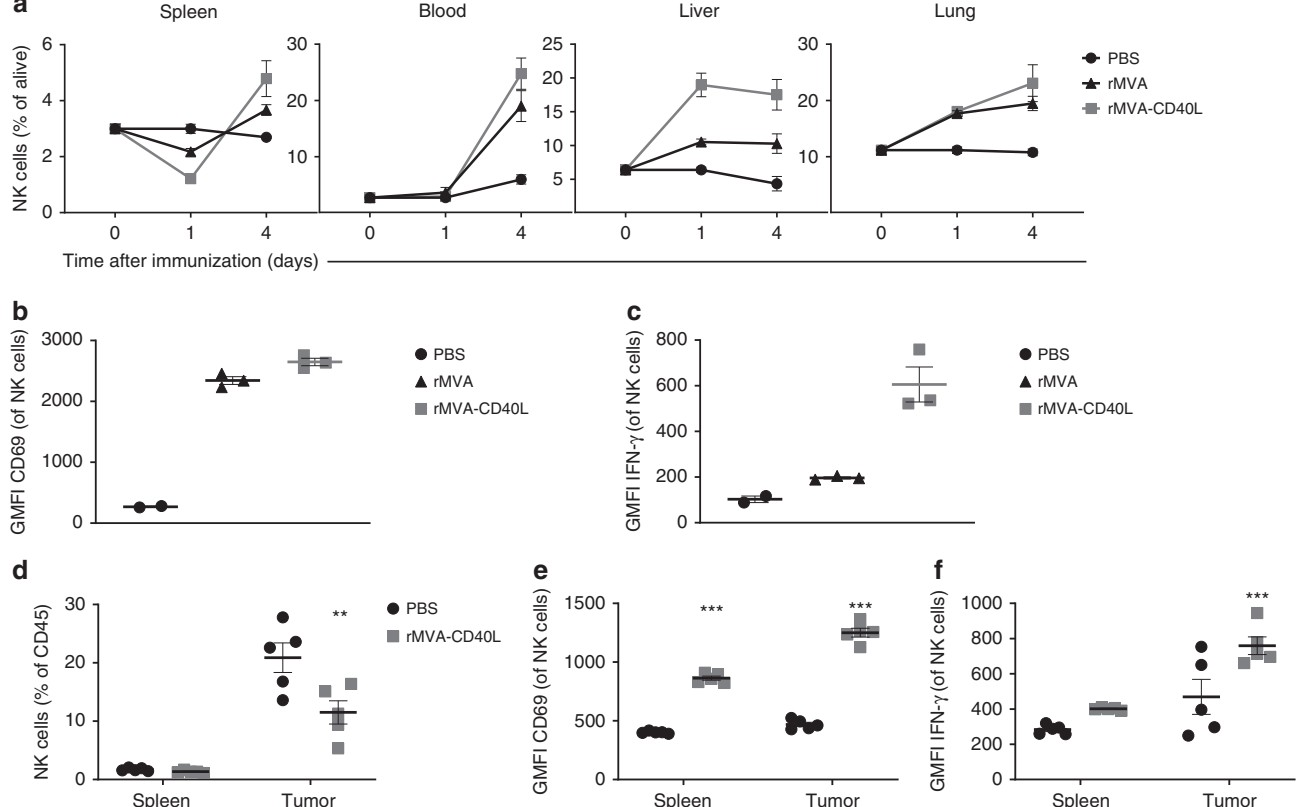

**Fig. 4** Strong NK cell activation and functionality upon systemic rMVA-CD40L immunization. **a** Systemic mobilization of NK cells upon intravenous rMVA immunization. C57BL/6 mice received PBS ($n = 2$ mice) or were immunized i.v. with $5 \times 10^7$ TCID$_{50}$ of either rMVA or rMVA-CD40L ($n = 3$ mice/group). Mice were killed at days 1 or 4 after immunization and the presence of NK cells (defined as CD3$^-$ NKp46$^+$ cells) in the spleen, blood, liver, and lung was assessed. Data are expressed as mean ± SEM, representative of two independent experiments. **b**, **c** rMVA or rMVA-CD40L induction of NK cell activation. C57BL/6 mice received PBS ($n = 2$ mice) or were immunized i.v. with $5 \times 10^7$ TCID$_{50}$ of either rMVA or rMVA-CD40L ($n = 3$ mice/group). Mice were killed at day 1 after immunization and the expression of CD69 (**b**) and IFN-γ (**c**) on splenic NK cells was analyzed by flow cytometry. Data are expressed as mean ± SEM, representative of two independent experiments. **d**, **e** Activation of splenic- and tumor-infiltrating NK cells upon rMVA-CD40L immunization. B16.OVA tumor-bearing mice ($n = 5$ mice/group) received PBS or were immunized i.v. with $5 \times 10^7$ TCID$_{50}$ of rMVA-CD40L. Mice were killed at day 2 after immunization and the presence of NK cells (**d**) as well as the expression of CD69 (**e**) and IFN-γ (**f**) on NK cells infiltrating the spleen and tumor was assessed by flow cytometry. Data are expressed as mean ± SEM, representative of two independent experiments. Two-way ANOVA was performed on **a** and **d**–**f**. **$p < 0.01$; ***$p < 0.005$

(Supplementary Fig. 5A), suggesting a mobilization of NK cells to the liver and lungs.

By day 4, NK cell frequencies in immunized mice increased in the spleen, lung, and blood, while remaining stable in the liver. The increase in NK cell percentage by day 4 after immunization either with rMVA or rMVA-CD40L was accompanied by a significant increase of Ki67 expression by NK cells in the beforementioned organs (Supplementary Fig. 5A), reflecting vaccine-induced proliferation. Therefore, intravenous immunization using rMVA or rMVA-CD40L induces a systemic expansion of NK cells.

We further determined whether intravenous immunization would result in enhanced NK cell activation. Expression of CD69 in splenic NK cells (Fig. 4b) was increased 1 day after intravenous immunization. In addition, rMVA-CD40L-immunized mice showed significantly higher CD69 expression on NK cells than those from rMVA-immunized littermates. This finding was also observed in the liver and lung-infiltrating NK cells (Supplementary Fig. 5B). Based on this finding, we further addressed whether the enhanced NK cell activation also impacted NK cell function. Therefore, we measured IFN-γ expression by NK cells after immunization, as this cytokine is important for NK cell function[30]. Interestingly, IFN-γ production by NK cells in the

spleen (Fig. 4c), liver, and lung (Supplementary Fig. 5C) was significantly increased upon rMVA-CD40L immunization compared with immunization with PBS or rMVA.

To test whether rMVA-CD40L would enhance NK cell activation in the tumor microenvironment, melanoma-bearing mice were killed 2 days after immunization, to analyze splenic and tumor NK cell infiltration. Whereas upon rMVA-CD40L immunization the percentage of NK cells among tumor leukocytes was decreased compared with PBS treatment (Fig. 4d), the expression of CD69 and IFN-γ was significantly increased in tumor-infiltrating NK cells after immunization (Fig. 4e, f), suggesting that therapeutic administration of rMVA-CD40L promotes NK cell activation in the tumor.

**Cooperation of rMVA-CD40L and tumor-targeting antibodies.** Antibody-dependent cellular cytotoxicity (ADCC) or phagocytosis are innate immune defense mechanisms largely exploited in the clinic by mAbs directed against TAAs expressed in different cancers such as HER2 or CD20[31]. Therefore, we prompted to determine whether rMVA-CD40L immunization combined with an antibody against a TAA would increase antitumor efficacy in vivo. Anti-TRP-1 (TA99) is an antibody directed against the

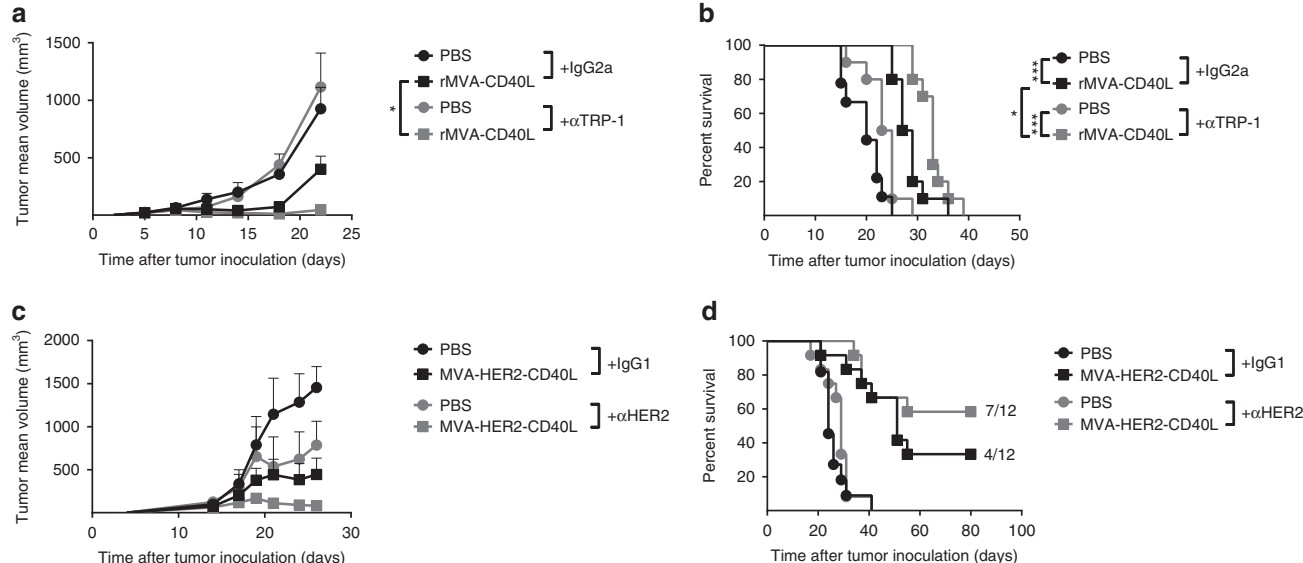

**Fig. 5** Combination of MVA-CD40L and tumor-targeting antibodies enhances tumor growth control. **a–d** Enhanced tumor growth control when rMVA-CD40L immunization is combined with tumor-targeting antibodies. **a**, **b** B16.OVA tumor-bearing mice received PBS or were immunized with $5 \times 10^7$ TCID$_{50}$ of rMVA-CD40L (Day 0). Mice received 200 μg of either IgG2a or anti-TRP-1 antibody i.p. at days −2, 2, 6, and 10. **a** Tumor size follow-up ($n = 5$ mice/group) and **b** overall survival. **c**, **d** Balb/c mice bearing 85–100 mm$^3$ CT26.HER2 received PBS or were immunized with $5 \times 10^7$ TCID$_{50}$ of MVA-HER2-CD40L (Day 0). Mice received 5 μg of either IgG1 or anti-HER2 antibody i.p. at days −2, 1, and 4. **c** Tumor size follow-up ($n = 5$ mice/group) and **d** overall survival. Data in **a** and **c** are expressed as mean ± SEM, representative of at least two independent experiments. One-way ANOVA was performed on **a**. Log-rank test on mouse survival was performed for **b**. *$p < 0.05$; **$p < 0.01$; ***$p < 0.005$

gp75 surface protein of melanoma cells and has shown therapeutic effects in early treatment regimens[32]. The combination of anti-TRP1 with single systemic rMVA-CD40L immunization of mice bearing established melanomas resulted in increased tumor growth control (Fig. 5a) and survival (Fig. 5b). Trastuzumab®, the first Food and Drug Administration (FDA)-approved targeted therapy for breast cancer, is a humanized IgG1 mAb that targets the HER2 receptor, known to induce ADCC in cells with high HER2 expression[33], among other mechanism of action. As human IgG binds mouse FcγR with similar affinity to that of mouse IgG[34,35], and human IgG1 induces ADCC by mouse NK cells and macrophages[34], we combined anti-HER2 treatment with MVA-HER2-CD40L in large CT26.HER2 tumors of volumes above 85–100 mm$^3$. Interestingly, a combination of anti-HER2 and systemic MVA-HER2-CD40L had a potent synergistic effect (Fig. 5c), resulting in tumor regression in 7 out of 12 treated mice (Fig. 5d). Hence, the combination treatment of systemic rMVA-CD40L immunization and TAA-specific antibodies such as anti-TRP-1 and anti-HER2 improves tumor growth control compared with monotherapy.

**Molecular and cellular components of the combined treatment**. TAA-specific mAb have been reported not only to activate innate immune cells expressing Fc receptors such as NK cells[36] or phagocytes[32,37] but also to stimulate cellular responses through enhancement of antigen cross-presentation[31,38,39]. To test for these various FcR-mediated effects, we used mice devoid of Fcγ receptors in a therapeutic setting against melanoma. Whereas *FcγR*-deficient tumor-bearing mice responded to systemic rMVA-CD40L immunization (Supplementary Fig. 6A, B), these mice did not benefit from the combination with anti-TRP-1 mAb neither in tumor growth (Fig. 6a) nor in survival (Fig. 6b). This result suggests the involvement of FcγR-proficient cells, including neutrophils, monocytes, macrophages, and NK cells, in the combination of anti-TRP-1 mAb and systemic rMVA-CD40L immunization. Further studies were conducted to dissect the role

of NK cells in the therapeutic efficacy of rMVA-CD40L. Mice deficient in IL15 receptor-α ($IL15r\alpha^{-/-}$) present a drastic reduction in the number of NK cells[40]. Indeed, peripheral blood NK cell frequencies were drastically reduced in $IL15r\alpha^{-/-}$ tumor bearers (Supplementary Fig. 7A), whereas transgene-specific and vector-specific CD8$^+$ T cells were expanded upon vaccination (Supplementary Fig. 7B, C, respectively). rMVA-CD40L immunization induced tumor growth control equally in wild-type (WT) and in $IL15r\alpha$-deficient tumor-bearing littermates (Supplementary Fig. 6C), and hence no differences in mouse survival between genotypes upon single immunization were observed (Supplementary Fig. 6D). However, the synergistic effect of rMVA-CD40L and anti-TRP-1 TAA-specific mAb was lost in $IL15r\alpha^{-/-}$ tumor-bearing mice (Fig. 6c, d), in contrast to the effects observed in WT counterparts treated with the combination.

Altogether, antitumor efficacy of single rMVA-CD40L immunization results in systemic NK and CD8$^+$ T cell activation and expansion. The combination treatment of systemic rMVA-CD40L immunization and TAA-specific antibodies such as anti-TRP-1 and anti-HER2 significantly improves tumor growth control. This synergistic effect depends on Fcγ receptors and NK cells.

**Discussion**
The recent FDA approval in 2017 of two CAR T cell-based therapies, Yescarta® and Kymriah®, emphasizes the importance of target recognition by T cells to achieve immune responses against cancer. Similarly, therapeutic cancer vaccines are designed to enhance tumor antigen presentation by APCs and to generate and increase tumor antigen-specific T cell responses. Due to their excellent adjuvant effects, viral vectors have been utilized as cancer vaccine platforms to promote maximal T cell expansion and differentiation, while exploiting their unique ability to activate the innate immune compartment. Among viral vector candidates, not only intrinsic differences in their genetic material, ability to replicate, or genetic adjuvantation but also extrinsic

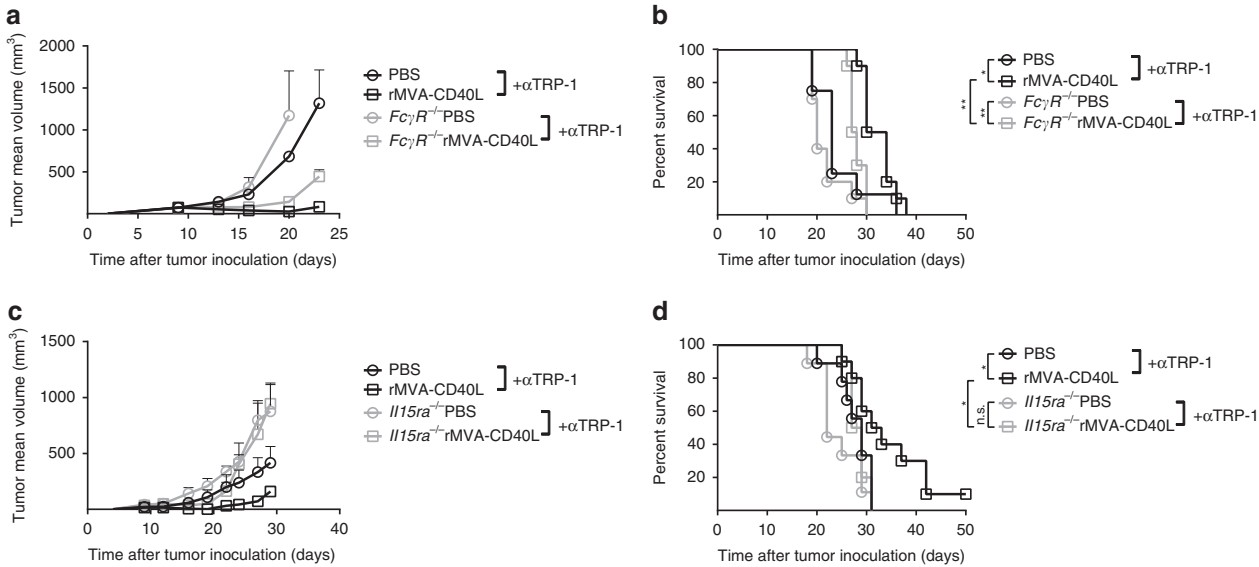

**Fig. 6** rMVA-CD40L/TAA mAb combination is dependent on Fcγ receptors and NK cells. **a**, **b** B16.OVA tumor-bearing wild-type and $Fc\gamma R^{-/-}$ mice were grouped according to tumor size. Tumor-bearing littermates either received PBS or were immunized with $5 \times 10^7$ TCID$_{50}$ of rMVA-CD40L (Day 0). Mice received 200 μg of anti-TRP-1 antibody i.p. at days −2, 2, 6, and 10. **a** Tumor size follow-up ($n = 5$ mice/group) and **b** overall survival ($n = 10$ mice/group). **c**, **d** B16.OVA tumor-bearing wild-type and $Il15ra^{-/-}$ mice were grouped according to tumor size. Tumor-bearing mice either received PBS or were immunized with $5 \times 10^7$ TCID$_{50}$ of rMVA-CD40L (Day 0). Mice received 200 μg of anti-TRP-1 antibody i.p. at days −2, 2, 6, and 10. **c** Tumor size follow-up ($n = 5$ mice/group) and **d** overall survival ($n = 10$ mice/group). **b**, **d** Overall survival of two merged independent experiments. Data in **a** and **c** are expressed as mean ± SEM. One-way ANOVA was performed on **a** and **c**. Log-rank test on mouse survival was performed for **b** and **d**. *$p < 0.05$; **$p < 0.01$

factors such as route of immunization or virus inactivation shape their therapeutic efficacy.

In the present study, a single intravenous immunization of mice bearing established tumors with rMVA promoted tumor growth control in both melanoma and lymphoma models. These results are in line with those of Fend et al.[14], where mice bearing orthotopic renal tumors received non-adjuvanted MVA-MUC1 intravenously at days 1 and 8 after tumor inoculation, resulting in little benefit in survival. Data from our group have shown that adjuvantation of rMVA by encoding CD40L strongly enhanced DC maturation and cytokine production, resulting in high and efficient cytotoxic T lymphocyte responses when given intravenously[10]. Viral platforms based on other DNA viruses such as non-replicating and replicating adenovirus expressing CD40L are mainly administered intratumorally several times[41–43]. Here we showed that a single systemic injection of rMVA adjuvanted with CD40L significantly improved tumor growth control compared with rMVA, leading to tumor rejection in multiple tumor models. More importantly, using three different antigen systems, we showed that the antitumor effects exerted by rMVA-CD40L were independent of the choice of antigen. rMVA-CD40L-mediated tumor growth control was achieved not only when using exogenous model antigens such as OVA or human antigens as HER2 encoded by the vaccine, but also when expressing either the endogenous retroviral elements p15E or AH-1A5 in WT tumor models lacking exogenous tumor antigens.

One possible explanation for the superior antitumor control by rMVA-CD40L might lie in the systemic administration of the vector. Intratumoral administration of CD40L-expressing vectors induces local priming of immune responses along with the release of alarmins that might help to strengthen immune activation in the case of oncolytic adenoviruses[42]. Intravenous immunization using rMVA-CD40L results in the infection of APCs in organs such as the spleen, lung, and liver[10,44]. Therefore, adjuvantation of rMVA with CD40L increases the likelihood of matured APC–T cell interactions to occur and thus generates strong systemic, Ag-specific T cell expansion. Another possible explanation is the

induction of humoral responses, as the vectors used in this study express tumor antigens and CD40/CD40L interactions are important in controlling B-cell immunity. Although B cell depletion led to a drastic reduction of circulating B cells prior to treatment (Supplementary Fig. 8A), it had no effect on the anti-tumor response (Supplementary Fig. 8B). In addition, MVA- and HER2-specific IgG in CT26.HER2 tumor-bearing mice (Supplementary Fig. 8D, E) had no impact on MVA-HER2-CD40L-induced antitumor effects (Supplementary Fig. 8B) and prolonged survival of the animals (Supplementary Fig. 8C). This result suggests that the role of B cells and B cell-induced antibodies is dispensable in the therapeutic effect of systemic rMVA-CD40L immunization.

rMVA and rMVA-CD40L immunization changed the phenotype of tumor-infiltrating CD4$^+$ and CD8$^+$ T cells. Exhausted T cells express multiple inhibitory receptors, including PD-1 and Lag3. Unexpectedly, we observed that rMVA-CD40L-induced, tumor-infiltrating antigen-specific CD8$^+$ T cells expressed significantly lower levels of both immune suppressive receptors. Systemic immunization using rMVA-CD40L induces, among other cytokines and chemokines, increased amounts of IL12p70 serum levels[10]. Both cytokines serve as signal 3 for antigen-dependent T cell activation[45]. IL12 but not IFN-α regulates the expression of PD-1 in CD8$^+$ T cells pre-conditioned for adoptive T cell transfer[43] and in HBV patients[46]. In addition, CD40L-induced IL12 plays a major role in the reinvigoration of CD8$^+$ T cells in chronic infection. Consistent with our data showing lowered PD-1 expression of CD4$^+$ and CD8$^+$ tumor-infiltrating T cells upon rMVA-CD40L treatment, Ngiow et al.[47] recently reported that agonistic CD40 antibody-induced IL12 was responsible for PD-1 downregulation on CD4$^+$ and CD8$^+$ T cells infiltrating murine mammary and renal carcinomas. However, contrary to our observations of additionally decreased Lag3 expression, upon CD40 agonistic antibody administration an increase in Lag3 expression in both TIL subsets was detected[47]. Previous results using subcutaneous immunization with MVA-BN-HER2 against CT26.HER2 tumors indeed showed a slight

increase of Lag3-positive cells in the tumor microenvironment 16 days after vaccination[48]. As Lag3 expression was significantly decreased to the same extent in CD4[+] and CD8[+] TILs by both rMVA and rMVA-CD40L systemic immunization regimes, we hypothesize that Lag3 downregulation on TILs in our model system is due to systemic priming by intravenous immunization with our viral vector platform.

Interestingly, both rMVA and rMVA-CD40L single immunizations decreased the percentage of FoxP3[+] $T_{reg}$ in the tumor microenvironment. This result is consistent with published data using subcutaneous immunization in a model of lung metastases[49], suggesting that rMVA-BN-based vectors decrease $T_{reg}$ frequencies within the tumor regardless of the route of immunization. In addition, PD-1 expression drastically dropped on tumor-infiltrating $T_{reg}$. Several lines of evidence point to the CD40/CD40L axis as crucial for $T_{reg}$ development and suppressive function in cancer[50,51]. Our data suggest that the tumor-infiltrating $T_{reg}$ decrease and loss of PD-1 expression observed is independent of genetic adjuvantation of rMVA.

Our results highlight the importance of CD8[+] T cell induction by rMVA-CD40L immunization in the observed antitumor response. As MVA has been shown to infect cDCs in vivo[23,52,53], based on our previous publication[10] we hypothesized that intravenous immunization using rMVA-CD40L could activate, among other immune cell populations, CD8α[+] cDCs that are specialized in antigen cross-presentation in the context of cancer[54], induce secretion of IL12p70, and thereby shape potent antitumor immune responses. To test this, we generated an rMVA expressing profilin, a *T. gondii*-derived protein that activates TLR11 and TLR12 exclusively in mouse CD8α[+] cDCs resulting in large amounts of IL12p70 secretion[25,55]. Our data show that both adjuvanted vectors rMVA-CD40L and rMVA-Profilin promote tumor growth control increased IL12p70 levels in sera, and antigen-specific CD8[+] T cell expansion to the same extent. Despite its lack of translational relevance, as TLR11 and TLR12 are not functionally expressed in humans, we believe that rMVA-Profilin constitutes a useful tool to depict the specific contribution of CD8α[+] cDCs for rMVA in mouse models of disease. As MVA infection modulates multiple innate cell subsets such as cDCs, plasmacytoid DCs, macrophages, and NK cells[44,56–58], additional studies would be needed to better address the specific contribution of antigen cross-presenting CD8α[+] cDCs in the context of generation of antitumor immune responses.

NK cells constitute an important arm of the innate immune system in poising a fast response to kill infected cells and tumor cells. Systemic infection using different replicating virus strains is described to induce NK cell proliferation and trafficking not only at the infected sites but also to other organs such as the spleen, liver, and lung in a cytokine-dependent manner[59–61]. Our data reflect NK cell mobilization in the spleen, liver, and lung 1 day after intravenous administration of either rMVA or rMVA-CD40L. In addition, rMVA immunization promoted upregulation of CD69 and IFN-γ expression by NK cells in the beforementioned organs, being significantly enhanced by rMVA-CD40L, most likely due to the increased IL12p70 production[62]. Indeed, IL12 has been reported to induce IFN-γ secretion by human NK cells in vitro[63], as well as to be necessary to suppress lung metastases in a NK cell-dependent manner[64]. We envision that systemic rMVA-CD40L-induced NK cell mobilization and activation would be of use in the context of metastatic disease, where NK cell activation has been proven to be key to achieve tumor regression in hepatic[65] and lung metastases models[64]. Consistent with this line of evidence, we observed significant tumor-infiltrating NK cell activation 2 days after rMVA-CD40L immunization, a time point in which intense CD8[+] T cell accumulation in the tumor microenvironment did not yet occur.

Guided by our results, we hypothesized that systemic NK cell expansion and activation upon MVA infection could be utilized to profit from NK lytic capacities that strengthen antitumor immune responses. This approach mimics a "two punch" strategy: rMVA-CD40L systemic immunization would activate APCs, leading to the secretion of pro-inflammatory cytokines[10], thereby enhancing innate immune cell activation and function, and afterwards priming potent antitumor CD8[+] T-cell responses. We report that rMVA immunization significantly enhanced NK cell activation and expansion. A wide array of therapeutic strategies based on the delivery of cytokines, immunostimulatory antibodies, or autologous NK cell infusions are being currently evaluated to enhance ADCC against clinically tested antitumor mAbs (rituximab®, cetuximab®, duratumumab®, or trastuzumab®, among others)[66]. Our data illustrate that rMVA-CD40L-mediated antitumor responses against established melanomas and aggressive HER2-expressing colon carcinomas are significantly improved when combined with cytotoxic mAbs anti-TRP-1 or anti-HER2 in vivo. A common function of all mAbs is the interaction of the mAb Fc domain with Fcγ receptors present in immune effector leukocytes[67]. Vaccination along with anti-TRP-1 mAb in the context of Fcγ receptor deficiency did not translate in additional therapeutic benefit, consistent with a previous report[32]. In addition, in the absence of NK cells, the improvement of therapeutic efficacy of combining anti-TRP-1 mAb and systemic rMVA-CD40L immunization was impaired. Even though our data suggest that NK cells are involved in the enhancement of antibody mediated antitumor activity, we cannot conclude that NK cells directly perform ADCC in our models. Another possible explanation could be that NK cell-dependent activation, e.g., via IFN-γ secretion, of other immune cells such as neutrophils or macrophages leads to enhanced FcR-mediated killing of antibody-coated tumor cells. Therefore, our data reflect the importance of both molecular and cellular components, namely Fcγ receptors and NK cells, respectively, in the therapeutic combination of anti-TAA mAbs and our vaccine platform. The latter results emphasize how the induction of antiviral innate immunity by systemic viral immunization can be employed to positively impact the responses to therapeutic vaccination.

In summary, our work here depicts the antitumor efficacy of systemic immunization with a non-replicating MVA encoding CD40L in different tumor models and mouse strains. We describe the expansion of antigen-specific tumor-infiltrating CD8[+] T cells and efficient IL12 production. In addition, we increased therapeutic efficacy in combination settings with TAA mAbs in a FcγR-dependent manner and demonstrated systemic NK cell activation and function upon rMVA-CD40L immunization. The synergistic antitumor effects observed after rMVA-CD40L + TAA mAb combination treatment supports the evaluation of this concept for clinical use.

## Methods

**Ethics statement**. Animal experiments were approved by the animal ethics committee of the government of Upper Bavaria (Regierung von Oberbayern, Sachgebiet 54, Tierschutz) and were carried out in accordance with the approved guidelines for animal experiments at Bavarian Nordic GmbH. When indicated, an experiment was conducted at Charles River Discovery Services (Morrisville, NC, USA) in compliance with the Association for Assessment and Accreditation of Laboratory Animal Care International.

**Mice and tumor cell lines**. Six- to 8-week-old female C57BL/6J (H-2[b]) and Balb/cJ (H-2[d]) mice were purchased from Janvier Labs. *Fcgr*[−/−] and *Il15ra*[−/−] mice were obtained from the University of Zürich. All mice were handled, fed, bred, and maintained either in the animal facilities at Bavarian Nordic GmbH or at the University of Zürich according to institutional guidelines.

CT26 murine colon carcinoma cell line expressing human HER2 (CT26.HER2) was licensed from the Regents of the University of California[48]. The B16.OVA melanoma cell line was a kind gift of Roman Spörri (University of Zürich).

EG7-OVA (ATCC® CRL-2113™) and CT26.WT (ATCC® CRL-2638™) cell lines were purchased from American Type Culture Collection (ATCC). MC38.WT colon carcinoma cells were the property of and were used at Charles River Discovery Services. Tumor cells were cultured in DMEM Glutamax medium supplemented with 10% fetal calf serum (FCS), NEAA, sodium pyruvate, and penicillin/streptomycin (all reagents from Gibco) in an incubator at 37 °C 5% $CO_2$. G418 (Gibco; 0.5 mg/ml) was added to EG7.OVA cultures according to ATCC instructions. All tumor cell lines used in experiments conducted at Bavarian Nordic were regularly tested for *Mycoplasma* by PCR (results available upon request). Briefly, mice were injected subcutaneously in the flank with $5 \times 10^5$ tumor cells. Regarding B16.OVA, prior to injection cells were admixed with 7 mg/ml Matrigel (Trevigen). Tumor diameter was measured at regular intervals using a caliper twice a week. Tumor volume was calculated using the formula: $V = (length \times width^2)/2$. Regarding the MC38.WT experiment performed at Charles River Discovery Services (Morrisville, NC, USA), 18 days after tumor inoculation, animals were grouped and data are displayed as time after tumor treatment.

**Generation of MVA-BN recombinants**. MVA-BN® was developed by Bavarian Nordic and is deposited at the European Collection of Cell Cultures (V00083008). All recombinants were generated from a cloned version of MVA-BN® in a bacterial artificial chromosome. The generation of a recombinant MVA-expressing OVA (MVA-OVA) and a MVA-OVA adjuvanted with murine CD40L (MVA-OVA-CD40L) was described in a previous study[10]. These viruses are herein referred to as rMVA and rMVA-CD40L, respectively. The gene encoding for the *T. gondii* profilin was synthetized (Geneart, Life technologies) and cloned into the MVA-OVA genome to generate MVA-OVA-Profilin, herein referred to as rMVA-Profilin. DNA encoding for a string of tumor-associated epitopes consisting of one melanoma-associated Trp2-derived epitope (SVYDFFVWL, H2-K^b) and two murine leukemia virus gp70-derived CD8^+ T cell epitopes, p15E (KSPWFTTL, H2-K^b) and the modified AH1[68], AH-1A5 (SPSYAYHQF, H2-L^d), each one separated by a five amino acid linker (ADARY) to promote correct epitope processing and presentation, were synthetized and are referred to as the TAA. The TAA and murine CD40L sequences were cloned into the MVA genome to generate MVA-TAA-CD40L. The genes encoding for the murine Twist-related protein 1, a transcription factor involved in epithelial to mesenchymal transition[69], and for a modified form of the HER2 were synthetized (Geneart) and cloned together with or without the gene for murine CD40L into the MVA genome to generate either rMVA-Twist-HER2 or rMVA-Twist-HER2-CD40L. For simplicity, the before-mentioned viruses will be referred to as either MVA-HER2 or MVA-HER2-CD40L. Infectious viruses were reconstituted from bacterial artificial chromosomes by transfecting bacterial artificial chromosome DNA into BHK-21 cells and superinfecting them with Shope fibroma virus as helper virus. After three additional passages of primary embryo fibroblasts, helper virus free MVA recombinant viruses were obtained. All viruses used in animal experiments were purified twice through a sucrose cushion.

**Immunizations**. Intravenous injections were given into a lateral tail vein with a total volume of 100 µl containing $5 \times 10^7$ TCID$_{50}$ of the respective MVA recombinants. When indicated, sera were collected 6 h after immunization for quantitative cytokine determination. When indicated, blood was collected 7 days after immunization for peripheral blood immune cell phenotyping.

**In vivo antibody treatment**. For CD8^+ T cell depletion, tumor-bearing littermates were inoculated intraperitoneally (i.p.) with 200 µg of rat anti-mouse CD8α (clone 2.43, BioXCell) at days −2, 2, 6, and 10 after immunization. Two hundred micrograms of rat IgG2b (clone LTF2, BioXCell) was used as immunoglobulin isotype control for CD8^+ T cell depletion. For B cell depletion, Balb/c mice were injected intravenously with 250 µg of rat anti-mouse CD20 (SA271G2, Biolegend) according to manufacturer's instructions 4 days before and 9 days after CT26. HER2 tumor cell subcutaneous inoculation. Two hundred micrograms of rat IgG2b (clone LTF2, BioXCell) was used as immunoglobulin isotype control. For anti-TRP-1 experiments, 200 µg anti-mouse/human TRP-1 (clone TA99) or rat IgG2a (Clone 2A3, both BioXCell) were injected i.p. at days −3, 0, 3, 6, and 10 after immunization. For anti-HER2 experiments, 5 µg anti-HER2-Trastuzumab-hIgG1 or hIgG1 (Invivogen) were injected i.p. at days −2, 1, and 4 after immunization

**Cell isolation**. When indicated, the spleen, liver, lung, and tumor were collectedfrom mice and incubated with 0.1 mg collagenase/DNase (Roche) for 30 min at 37 °C. Single-cell suspensions were prepared by mechanically disrupting the organs through a 70 µm cell strainer (Falcon). Tumor-infiltrating mononuclear cells, liver cells, and lung cells were isolated by centrifugation with 44% Percoll (GE Healthcare). Cells were then subjected to red blood cell lysis (Sigma-Aldrich). Blood was collected in PBS containing 2% FCS, 0.1% sodium azide, and 2.5 U/ml heparin. Peripheral blood mononuclear cells (PBMCs) were prepared by lysing erythrocytes with red blood cell lysis buffer. Mononuclear cells from the above-mentioned organs were washed, resuspended in RPMI + 2% FCS (Gibco), counted and kept on ice until further analysis.

**Flow cytometry**. Tissue mononuclear cell suspensions were stained for 30 min at 4 °C in the dark using fixable live/dead staining kits according to manufacturer's instructions prior to staining (AmCyan, Life Technologies). When indicated, tissue or blood cell suspensions were stained using H-2K^b dextramers loaded either with OVA$_{257–264}$-peptide (SIINFEKL) or with B8R$_{20–27}$ (TSYKFESV) according to manufacturer's instructions (Immudex). Antibodies against CD3 (17A2, BV605 1:100 or 145–2C11, BV421 1:200), CD4 (RM4-5, BV650 1:200), CD8α (53^−6.7, BV785 1:100), CD19 (1D3, PE Cy7 1:1000), CD44 (IM7, APC eFluor780 1:100), CD45 (30-F11, FITC 1:1600 or PE Cy7, 1:3200), CD45R/B220 (RA3-6B2, PerCP Cy5.5 1:200), CD69 (H1.2F3, BV421 1:100), CD70 (FR70, PerCP eFluor710 1:25), CD223/Lag3 (C9B7W, PerCP eFluor710 1:50), CD279/PD-1 (29F.1A12, BV605 1:100 or RMP1-30, APC 1:25), CD335/NKp46 (29A1.4, PerCP eFluor710 1:25), Ki67 (35/Ki67, FITC 10 µl/sample), IFN-γ (XMG1.2, PE Cy7 1:200), FoxP3 (FJK-16s, APC 1:20), and MHC-II (M5/114.15.2, PerCP eFluor780 1:200) were purchased from BD Biosciences, Thermo Fisher, or Biolegend. For FoxP3 transcription factor and Ki67 staining cells were fixed using FoxP3 Staining Kit (Thermo Fisher). For intracellular IFN-γ cytokine staining, cell suspensions were stained and fixed for intracellular cytokine detection using IC Fixation & Permeabilization Staining kit (Thermo Fisher). All cells were acquired using a digital flow cytometer (LSR II, BD Biosciences) and data were analyzed with FlowJo software version 10.3 (Tree Star).

**Cytokine detection**. Cytokine concentrations in serum were determined by Multiplex Luminex assays according to manufacturer's instructions (Thermo Fisher). Analysis was performed using Masterplex 2010 version 2.0.0.77 (Hitachi Solutions, Ltd).

**Enzyme-linked immunosorbent assay**. To detect MVA IgG antibodies, MVA antigen was coated onto microtiter plates (Corning) at 1:500 (100 µl/well) in PBS 200 mM Na$_2$CO$_3$ overnight at 4 °C. The following day, plates were washed in PBS (Gibco)−0.05% Tween 20 (Sigma-Aldrich) (T-TBS) and were blocked with 200 µl/well PBS–5% FCS–0.05% Tween 20 for 2 h at room temperature (RT). Plates were then washed and air dried. Sera were diluted to 1:100 in blocking buffer and incubated for 1 h at RT, then washed again, and goat anti-mouse IgG-horseradish peroxidase (HRP) conjugate (Serotec/BioRad) diluted 1:5000 in blocking buffer was added to the wells and incubated for 1 h at RT. After washing in T-PBS, plates were developed by adding 50 µl/well TMB peroxidase substrate (Sigma) for 30 min in the dark. Development was stopped by the addition of H$_2$SO$_4$ 1N and OD read at 450 nm in a Sunrise FC plate reader (TECAN).

To detect HER2 IgG antibodies, goat anti-human IgG Fc Fragment (Jackson Immuno Research) was used to coat microtiter plates 1:500 (50 µl/well) in PBS 200 mM Na$_2$CO$_3$ overnight at 4 °C. The following day, plates were washed in T-PBS and blocked with 200 µl/well RPMI–10% FCS for 1 h at 37 °C 5% $CO_2$. Then, recombinant human ErbB2/Fc Chimera (R&D Systems) was plated at 40 ng/ml (100 µl/well) in RPMI- 10% FCS for 2 h at RT. Plates were then washed and air dried. Sera were diluted to 1:5000 in blocking buffer and incubated for 2 h at RT, then washed in T-PBS again, and sheep anti-mouse IgG H + L HRP conjugate (BioRad) diluted 1:10,000 in PBS + 1% bovine serum albumin was added to the plate for 2 h at RT. After washing in T-PBS, plates were developed by adding 50 µl/well TMB peroxidase substrate (Sigma) for 10 min in the dark. Development was stopped by the addition of H$_2$SO$_4$ 1N and OD read at 450 nm in a Sunrise FC plate reader (TECAN).

**Statistical analysis**. Statistical analyses were performed as described in the figure legends using GraphPad Prism version 7.02 for Windows (GraphPad Software, La Jolla, CA). For immunological data, data shown are the mean and SEM. Either analysis of variance with multiple comparisons test or one-tailed unpaired Student's $t$-tests were used to determine statistical significance between treatment groups. For tumor-bearing mice survival after treatment, Log-rank tests were performed to determine statistical significance between treatment groups.

**Reporting summary**. Further information on research design is available in the Nature Research Reporting Summary linked to this article.

## Data availability
The authors declare that the data supporting the findings of this study are available within the paper and its Supplementary Information files or available from the authors upon reasonable request.

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

## Acknowledgements

We thank Christian Krause, Yvonne Terkowski, and Kerstin Zehentbauer for excellent technical support in handling the animal facility and Vaccine Generation for producing virus stocks.

## Author contributions

Experiment design: J.M.-E., M.H., H.H. and H.L. Virus generation: S.T.W. Experiment performance: J.M.-E., M.H., M.T., M.G., R.G., B.B., R.K., F.G., G.F. and M.S. Data analysis and interpretation: J.M.-E., M.H., H.H. and H.L. Manuscript writing and scientific discussions: J.M.-E., M.H., M.S., P.C., H.H. and H.L.

## Competing interests

The authors enlisted are or have been employees of Bavarian Nordic.
