## [Peer Review File · Nature Communications]

Reviewers' comments:

Reviewer #1 (Remarks to the Author):

The manuscript by Medina-Echeverz et al. describes the use of a tumor-vaccine strategy based on systemic administration of a vaccinia (MVA) vector co-expressing a tumor associated-antigen (TAA) and CD40L, this last molecule able to stimulate dendritic cells (DCs), working as immunological "adjuvant". The authors show in several preclinical models that this strategy leads to better antitumor responses than those obtained with the same vector expressing only the tumor antigen. A detailed characterization of the immunological responses led to the conclusion that not only CD8⁺ T-cells and CD8 α ⁺ conventional DCs are required for efficacy, but also NK cells play an important role, being more expanded and activated by the vector expressing CD40L. Finally, as a possible mechanism for the role of NKs, they show that ADCC, most likely mediated by NK Fc engagement, could be favoring the elimination of tumor cells. In this sense, combination of the MVA vector and anti-TAA mAbs showed an improvement of the response.

Although an MVA vector expressing an antigen and "armed" with CD40L has been previously described by the authors in immunization studies (see ref. 9 in paper), its use for cancer treatment is novel and could represent an interesting strategy for cancer therapy, although it will be limited to tumors for which a tumor antigen is known.

Some major points that could strengthen the conclusions:

- Although it is remarkable that the authors could obtain antitumor responses by systemic administration of the vector, responses in some tumor models were rather moderate (B16.OVA & EG7.OVA). Could it be possible to enhance these responses with repetitive administrations of the vector?

- The authors analyzed in detail cellular immune responses induced by the MVA-CD40L vector (CD8 T cells, DCs and NKs). However, the fact that the vector expresses a tumor antigen and that ADCC could be involved in its antitumor activity (Fig. 5) suggests that humoral responses could also play an important role. Have the authors analysed the titers of antibodies against tumor antigens (i.e. against OVA in the B16-OVA model). In this respect it would also be interesting to evaluate the effect of B cells depletion on the vector efficacy.

- Regarding the role of NK cells in this antitumor strategy, it would be interesting to show whether depletion of these cells with specific antibodies can inhibit the therapeutic efficacy of MVA-CD40L vectors.

- Regarding the combination of MVA-CD40L with anti-TAA mAbs, the effect mediated by Fc-gammaR on antitumor efficacy is not very clear (no significant differences observed in Fig. 5E & F between WT and Fc-gammaR^{-/-} mice). Could the authors try to demonstrate the role of ADCC in an indirect way? For example by using an anti-TAA mAb with an isotype not able to engage Fc-gamma receptor? (Note, clone TA99 anti TRP-1 is also commercially available with a Fc silent domain).

Minor points:

- The authors should explain clearly at the beginning of Results that they are using MVA expressing OVA or HER (for TAA is explained well). Although this is defined in M&M, the fact that they abbreviate the names of the vectors to only MVA or MVA-CD40L, can create some confusion.

- Fig 5C. It is indicated that control groups were treated with anti IgG1 while in Fig. 5D it says IgG2a. According to M&M the anti HER mAb is a human IgG1 mAb. Please correct, because both figures correspond to the same experiment. Please, also include some comment and reference showing that human IgG1 can engage mouse Fc receptors, inducing ADCC.

- Figs. 5E & F are too busy to distinguish well between the different groups. Maybe graphs could be splitted in two, showing in parallel graphs with isotype control IgG and specific anti-TAA mAb, respectively.
- Fig S4A. Indicate in the axis that you show NK-Ki67 positive cells
- Some references are not complete (i.e. 5 & 6)

Reviewer #2 (Remarks to the Author):

The authors investigate the immunostimulatory activity of a modified vaccinia virus Ankara (rMVA) vector containing a CD40L expression cassette in the setting of tumor immunity. The underlying concept is that the overexpression of CD40L may interact with CD40 expressed on professional antigen presenting cells, such as dendritic cells and license them for efficient activation of T cell responses. The current study represents a follow up of a previous study published six years ago, demonstrating that the MVA-CD40L system allows boosting cytotoxic T cell responses (Lauterbach et al., *Front. Immunol.* 2013). In the current study the authors use the B16.Ova, EG7.OVA, CT26, and CT26 expressing HER2 tumor models and infect tumor bearing mice with rMVA alone or with rMVA-CD40L. By using Batf3^{-/-} mice lacking CD8⁺ dendritic cells the authors show that the priming of T cells may be reduced. Expectedly, the immune cells controlling tumor growth are cytotoxic T cells. Moreover, less pro-inflammatory cytokines seem to be present in Batf3^{-/-} mice upon rMVA-CD40L infection. In addition to T cells the authors also investigate the effect of rMVA-CD40L on NK cell activity and suggest that ADCC activity may be enhanced.

Major experimental concerns:

The authors selectively chose very immunogenic tumor models either expressing OVA or HER2. In the only wildtype model they have included a tumor antigen in their MVA vector, which is not comparable to the other experimental settings. The impact of the study could be enhanced if the authors would compare their data to B16 and EG7 alone.

In Figure 1 the rMVA control is not included consistently. As rMVA alone shows strong effects in EG7.OVA for example, having this set of data for all other models would help the interpretation of the data. This concern is of relevance for the entire manuscript (see also later).

In Figure 3C it is quite evident that tumor growth is enhanced in Batf3^{-/-} at baseline. Thus, the most simple and straightforward explanation for the reduced effects of rMVA-CD40L is that the tumor size is too large to allow a complete control via T cells. In fact there seems to be a significant effect, which the authors do not test for statistical significance. The way the data is shown and discussed is misleading.

Further along these lines it has been demonstrated that Batf3 also plays a role in T cells. As long as the authors do not use a DC or T cell specific Batf3^{-/-} mouse line many of the drawn conclusions remain speculative.

In Figure 3E the authors show differences in cytokine levels. It remains unclear, however, if this is simply due to the MVA infection. Without the proper rMVA control infection the data is hard to interpret.

In Figure 5 the authors try to make the point that rMVA-CD40L may enhance ADCC via NK cells. However, the authors never show that NK cells contribute to the anti-tumor activity in their specific setting. The literature they cite and some other additional studies show that NK cells are

actually not involved in killing B16 tumors in vivo. Moreover, the critical parts of the Figure (5E and F) are hard to read and it seems that no significant differences could be detected.

Reviewers' comments:

Reviewer #1 (Remarks to the Author):

The manuscript by Medina-Echeverez et al. describes the use of a tumor-vaccine strategy based on systemic administration of a vaccinia (MVA) vector co-expressing a tumor associated-antigen (TAA) and CD40L, this last molecule able to stimulate dendritic cells (DCs), working as immunological “adjuvant”. The authors show in several preclinical models that this strategy leads to better antitumor responses than those obtained with the same vector expressing only the tumor antigen. A detailed characterization of the immunological responses led to the conclusion that not only CD8+ T-cells and CD8 α + conventional DCs are required for efficacy, but also NK cells play an important role, being more expanded and activated by the vector expressing CD40L. Finally, as a possible mechanism for the role of NKs, they show that ADCC, most likely mediated by NK Fc engagement, could be favoring the elimination of tumor cells. In this sense, combination of the MVA vector and anti-TAA mAbs showed an improvement of the response.

Although a MVA vector expressing an antigen and “armed” with CD40L has been previously described by the authors in immunization studies (see ref. 9 in paper), its use for cancer treatment is novel and could represent an interesting strategy for cancer therapy, although it will be limited to tumors for which a tumor antigen is known.

Some major points that could strengthen the conclusions:

- *Although it is remarkable that the authors could obtain antitumor responses by systemic administration of the vector, responses in some tumor models were rather moderate (B16.OVA & EG7.OVA). Could it be possible to enhance these responses with repetitive administrations of the vector?*

To address whether repetitive administration of our vaccines would increase therapeutic responses in a tumor model where the observed effects are rather moderate, we compared prime versus prime and boost IV immunizations of either rMVA or rMVA-CD40L in the B16.OVA tumor model.

Figure 1 Point-by-point Reply. Repeated intravenous administration of rMVA vectors did not result in increased therapeutic efficacy. Briefly, C57BL/6 mice received B16.OVA cells subcutaneously in the flank. Seven days later, when tumors were above 50 mm³, mice were grouped (n=5 mice/group) and treated intravenously either with PBS or immunized with 5x10⁷ TCID₅₀ of the mentioned rMVA viruses in the table (black dotted line; see table inserted). Seven days after prime immunization, tumor-bearing mice received a boost immunization with 5x10⁷ TCID₅₀ of the mentioned rMVA viruses (black

dashed line). Tumor growth was measured at regular intervals. Data are expressed as Mean \pm SEM. One-way ANOVA comparing MVA immunization vs PBS. * $P < 0.05$, *** $P < 0.005$.

Regarding rMVA prime boost (Figure 1, open triangle), we observed a mild delay in tumor growth which was not statistically significant compared to rMVA prime (closed triangle). No difference in antitumor effects was observed when rMVA-CD40L prime (closed square) versus prime boost (open square) were compared. We incorporated this result as Supplementary Figure 2 and added the following sentence to the *Results* section of the manuscript: "Repeated administration of rMVA-CD40L did not increase antitumor responses against B16.OVA melanoma tumors (Figure S2)."

- The authors analyzed in detail cellular immune responses induced by the MVA-CD40L vector (CD8 T cells, DCs and NKs). However, the fact that the vector expresses a tumor antigen and that ADCC could be involved in its antitumor activity (Fig. 5) suggests that humoral responses could also play an important role. Have the authors analysed the titers of antibodies against tumor antigens (i.e. against OVA in the B16-OVA model). In this respect, it would also be interesting to evaluate the effect of B cell depletion on the vector efficacy.

B cells are key elements in the generation of adaptive immune responses through the production of highly specific antibodies. To mimic a more clinically relevant setting to address the role of B cells and of antibody production upon systemic MVA immunization, we used the CT26.HER2 tumor model. 4 days before and 9 days after subcutaneous, CT26.HER2 tumor cell implantation, mice received either anti-CD20 antibody or the isotype control IgG2b IV, according to manufacturer's instructions. Pilot experiments performed in our lab using anti-CD20 antibody showed that IV administration resulted in a significant depletion of B cells in spleen, blood and tumor, but not in the bone marrow (data not shown). To determine the degree of B cell depletion, mice were bled 8 days after tumor inoculation.

Figure 2 Point-by-point Reply. Role of B cells on systemic cancer immunotherapy with MVA-HER2-CD40L. Balb/c mice received 5×10^5 CT26.HER2 cells subcutaneously. 13 days later, mice were grouped ($n=6$ mice/group) and IV injected either with PBS or immunized with 5×10^7 TCID₅₀ MVA-HER2 or 5×10^7 TCID₅₀ MVA-HER2-CD40L. Mice received 250 μ g anti-CD20 Ab (clone SA271G2) or IgG2b IV 4 days prior and 9 days after tumor cell inoculation. Blood for the detection of specific antibodies

was withdrawn 7 days before and 27 days after CT26.HER2 tumor cell inoculation. Tumor growth was measured at regular intervals. (A) Percentage of CD3⁺ CD19⁺ MHC-II⁺ CD45R⁺ cells of CD45⁺ cells in blood 8 days after CT26.HER2 cell inoculation; (B) Tumor mean volume; (C) Kaplan Meier plot showing survival of mice. ELISA on serum samples taken at day 27 after tumor inoculation against MVA IgG (D) or HER2 IgG (E). This experiment has been partially shown in Figures 1H and 1I Data expressed as Mean \pm SEM. B was analyzed using One-way ANOVA comparing rMVA immunization vs PBS; C was analyzed using Log-rank test. * $P < 0.05$, *** $P < 0.005$.

B cells in the blood were drastically diminished after anti-CD20 antibody treatment (Fig. 2A). Before therapeutic treatment of CT26.HER2 tumor-bearing mice using MVA variants, they received another dose of either anti-CD20 Ab or IgG2b Isotype control IV to ensure long term depletion.

CT26.HER2 tumors grew slower in mice devoid of B cells (Figure 2B). Consistent with previous data, MVA-HER2 immunization resulted in the delay of tumor growth compared to PBS controls in IgG2b treated mice (Figure 2B). Interestingly, CD20 depletion enhanced MVA-HER2 antitumor efficacy (Figures 2B and 2C). However, those changes were not significant when compared to IgG2b treated tumor-bearers. Importantly, the adjuvantation of MVA-HER2 with CD40L drastically increased the antitumor effect and led to complete tumor clearance in all treated animals (Figures 2B and 2C). However, B cell depletion in tumor-bearing mice did not interfere with MVA-HER2-CD40L-mediated anti-tumor responses.

To analyze whether B cell depletion had an impact on the generation of MVA and HER2-specific antibodies, MVA and HER2-specific IgG antibodies in sera were tested by ELISA. Consequently, IgG antibodies were strongly diminished in anti-CD20 Ab treated animals for both MVA and HER2 responses (Figures 2D and 2E.). Of note, in some samples we could still detect MVA and HER2-specific antibodies despite anti-CD20 treatment. However, we could correlate these samples to the mice that showed insufficient B cell depletion in the blood on day 8. Interestingly, the mere presence of the CT26.HER2 tumor was sufficient to induce high amounts of HER2-specific antibodies (Figure 2E); however, the antibody titers could not be further enhanced by systemic immunization either using MVA-HER2 or MVA-HER2-CD40L.

Taken together, this experiment shows that B cells do not play a major role for IV MVA-HER2-CD40L - mediated antitumor effects. In fact, they slightly promoted tumor growth, as tumors in B cell-depleted mice grew to some extent slower than the control groups. An explanation for this result could be the absence of regulatory B cells after depletion, which have been shown to inhibit antitumor immune responses by the expression of IL-10, IL-35 or TGF- β^1 . This observation needs to be taken carefully, since different systems to assess B cell contributions have been explored, reporting different outcomes.

The following sentence has been added to the *Discussion* section: "Another possible explanation is the induction of humoral responses, since the vectors used in this study express tumor antigens and CD40/CD40L interactions are important in controlling B cell immunity. Although B cell depletion led to a drastic reduction of circulating B cells prior to treatment (Figure S5A), it had no effect on the antitumor response. In addition, MVA – and HER2 -specific IgG in CT26.HER2 tumor bearing mice (Figure S5D and S5E) had no impact on MVA-HER2-CD40L -induced antitumor effects (Figure S5B) and prolonged survival of the animals (Figure S5C).".

- Regarding the role of NK cells in this antitumor strategy, it would be interesting to show whether depletion of these cells with specific antibodies can inhibit the therapeutic efficacy of MVA-CD40L vectors.

The data presented in this manuscript shows how systemic rMVA-CD40L immunization results in NK cell expansion and proliferation in different organs (Figure 4A and Supplementary Figure S4A). In addition, systemic rMVA-CD40L immunization results in IL12p70 production by *Batf3*-proficient DCs (Figure 3C), a key cytokine in the induction of NK cell activation, along with type I interferon and IL-18^{2,3,4}. In naïve mice depletion of NK cells using anti NK1.1 (clone PK136) is highly effective, however, we found that after systemic rMVA-CD40L immunization we only observed some reductions in NK cell percentages but no effective depletion (data not shown). Under this condition NK cells seem to be depletion resistant. In line with that, we have observed that the anti-apoptotic molecule Bcl-X_L was upregulated on NK cells in spleens and lungs 24 hours upon systemic rMVA-CD40L immunization (unpublished observations). Therefore, to address the role of NK cells in the therapeutic efficacy of rMVA-CD40L vectors, we took advantage of mice deficient in IL15R α (IL15R α ^{-/-}), where NK cell numbers are drastically reduced⁵. We thus designed an experiment to determine whether the synergy between TAA mAb treatment and systemic immunization with rMVA-CD40L against B16.OVA tumors is influenced by the presence of NK cells. We observed that tumors seemed to grow faster in IL15R α ^{-/-} compared to wild type tumor-bearers at later timepoints (Figure 3). Upon treatment with rMVA-CD40L, we observed that immunization induced tumor growth control both in WT and in IL15R α -deficient tumor-bearing littermates (Figures 3A and 3B). No significant differences in mouse survival were observed in the comparison between WT or IL15R α ^{-/-} tumor-bearing mice solely immunized with rMVA-CD40L (Figure 3B).

Figure 3 Point-by-point Reply. rMVA-CD40L/ anti-TRP-1 antitumor effects are dependent of IL15R alpha signalling. B16.OVA tumor-bearing wild type and *IL15R α ^{-/-}* mice were grouped according to tumor size. Tumor bearing mice either received PBS or were immunized with 5x10⁷ TCID₅₀ of rMVA-CD40L (Day 0). Mice received 200 μ g of anti-TRP-1 antibody i.p. at days -2,2,6 and 10; (A and C) Tumor size follow-up (n= 5 mice/group) according to combination either with IgG2a (A) or with anti-TRP-1 mAb (C); (B and D) overall survival (n=10 mice/group) of mice according to combination either with IgG2a

(B) or with anti-TRP-1 mAb (D). Data in A and C are representative of two independent experiments and are expressed as Mean \pm SEM. B and D represent overall survival of two merged independent experiments. One-way ANOVA was performed on figures A and C. Log rank test on mouse survival was performed for figures B and D. *, $P < 0.05$; **, $P < 0.01$.

Interestingly, the synergistic effect of rMVA-CD40L/anti-TRP-1 was lost in *IL15 α ^{-/-}* mice (Figures 3C and 3D), in contrast to the enhanced tumor growth control observed in wild type mice treated with the combination, resulting in increased mouse survival.

We investigated the presence of NK cells as well as the occurrence of antigen-specific CD8⁺T cells in peripheral blood one week after intravenous vaccination, since IL15 receptor alpha expression is induced in CD8⁺T cells upon activation⁶. Peripheral blood NK cell frequencies were drastically reduced in *IL15 α ^{-/-}* mice (Figure 4A). OVA-specific and vector-specific CD8⁺T cells were expanded upon vaccination in *IL15 α* -deficient mice (Figures 4B and 4C respectively).

Figure 4 Point-by-point Reply. Supplementary Figure S7 in the manuscript, showing that rMVA-CD40L expands antigen-specific CD8⁺T cells in the absence of IL15 α . WT and *IL15 α ^{-/-}* B16.OVA tumor bearing mice (n=5 mice/group) were treated as described above. Seven days after immunization, mice were bled and peripheral blood analysis of immune cell subsets by flow cytometry was conducted. A) Percentage of NK cells-gated as CD3⁺NKp46⁺ in peripheral blood; B) Frequency of OVA-specific CD8⁺T cells in peripheral blood; C) Frequency of MVA-specific CD8⁺T cells in peripheral blood. Data in A, B and C are representative of two independent experiments. Data expressed as Mean \pm SEM. One-way ANOVA. * represents the statistically significant differences among treatment groups from the same genotype, whereas # represents statistically significant differences among the same treatment group compared between wild type and *IL15 α ^{-/-}* littermates. ### $P < 0.01$, ### $P < 0.05$, *** $P < 0.005$.

We have shown in this experiment that the absence of NK cells does not affect the therapeutic efficacy of intravenous rMVA-CD40L *per se*. rMVA-CD40L immunization resulted in significant expansion of antigen-specific CD8⁺T cells in *IL15 α* knockout tumor-bearers, which are critical for our vaccine-induced antitumor immune response as shown in the manuscript (Figure 3 Manuscript). However, NK cells seem to be important when our vaccine is combined with TAA mAbs, being the synergistic effect of the treatment lost in the absence of NK cells. This result is in line to the one obtained using *Fc γ R* knockout mice shown in Figure 5. Altogether, the manuscript best describes the molecular and cellular components that play key roles in the combination of rMVA-CD40L with TAA-specific mAbs.

The following paragraphs have been added to the *Results* section:” Further studies were conducted to dissect the role of NK cells in the therapeutic efficacy of rMVA-CD40L. Mice deficient in IL15 receptor alpha (*IL15 α ^{-/-}*) present a drastic reduction in the number of NK cells⁵. Indeed, peripheral blood NK cell frequencies were drastically reduced in *IL15 α ^{-/-}* tumor bearers (Supplementary Figure S7A), whereas transgene-specific and vector specific CD8⁺T cells were expanded upon vaccination

(Supplementary Figures S7B and S7C respectively). rMVA-CD40L immunization induced tumor growth control equally in WT and in *IL15R α* -deficient tumor-bearing littermates (Figure S6C), and hence no differences in mouse survival between genotypes upon single immunization were observed (Figure S6D). However, the synergistic effect of rMVA-CD40L and anti-TRP-1 TAA-specific mAb was lost in *IL15R α* ^{-/-} tumor-bearing mice (Figures 6C and 6D), in contrast to the effects observed in wild type counterparts treated with the combination. “

- Regarding the combination of MVA-CD40L with anti-TAA mAbs, the effect mediated by Fc-gammaR on antitumor efficacy is not very clear (no significant differences observed in Fig. 5E & F between WT and Fc-gammaR^{-/-} mice). Could the authors try to demonstrate the role of ADCC in an indirect way? For example by using an anti-TAA mAb with an isotype not able to engage Fc-gamma receptor? (Note, clone TA99 anti-TRP-1 is also commercially available with a Fc silent domain).

We would like to thank both reviewers with their suggestions in this regard. To bring clarity to the presentation of our data, we decided to divide the presented information as follows.

In Figure 5, we are showing that combining intravenous vaccination of rMVA-CD40L with TAA mAbs results in enhanced tumor growth control when compared to single treatments. Regarding the increased survival when combining MVA-HER2-CD40L with anti HER2, we included the number of tumor-free mice upon treatment (see below).

Figure 5 Point-by-point Reply. Revision of Figure 5 in the manuscript, showing how combination of MVA-CD40L and tumor-targeting antibodies enhances tumor growth control (A-D) Enhanced tumor growth control when rMVA-CD40L immunization is combined with tumor-targeting antibodies. (A, B) B16.OVA tumor bearing mice received PBS or were immunized with 5×10^7 TCID₅₀ of rMVA-CD40L (Day 0). Mice received 200 μ g of either IgG2a or anti-TRP-1 antibody i.p. at days -2,2,6 and 10; (A) Tumor size follow-up (n= 5 mice/group) and (B) overall survival. (C, D) Balb/c mice bearing 85-100 mm³ CT26.HER2 received PBS or were immunized with 5×10^7 TCID₅₀ of MVA-HER2-CD40L (Day 0). Mice received 5 μ g of either IgG1 or anti-HER2 antibody i.p. at days -2,1 and 4; (C) Tumor size follow-up (n= 5 mice/group) and (D) overall survival. Data in A and C are expressed as Mean \pm SEM, representative of at least two independent experiments. One-way ANOVA was performed on figures A and C. Log rank test on mouse survival was performed for figures D, F and H. *, P < 0.05; **, P < 0.01; ***, P < 0.005.

Then, we split the mechanism of action of the treatment combination into two figures. First, in the present Figure 6 (Supplementary Figure S7 in the manuscript), we show the combination of control immunoglobulin with systemic vaccination with rMVA-CD40L in *FcγR*^{-/-} and *IL15ra*^{-/-} phenotypes, since this information in WT mice has been shown beforehand in Manuscript Figures 3 and 5. Regarding tumor growth in *FcγR*^{-/-} and *IL15ra*^{-/-} tumor-bearers compared to wild type, we observed that tumors in the knockout mice grew slightly faster, but this difference was neither statistically significant for *FcγR*^{-/-} nor *IL15ra*^{-/-} mice. Referring to comparing intravenous rMVA-CD40L immunization, no significant differences in terms of survival among genotypes were observed when immunizations were not combined with tumor targeting antibodies.

Figure 6 Point-by-point Reply. Supplementary Figure S6 in the manuscript, showing that single, systemic immunization with rMVA-CD40L enhances tumor growth control both in *FcγR*^{-/-} and *IL15ra*^{-/-} tumor-bearing mice, (A,B) Performed along with Figures 6A and 6B. B16.OVA tumor-bearing wild type and *FcγR*^{-/-} mice were grouped according to tumor size. Tumor-bearing mice either received PBS or were immunized with 5x10⁷ TCID₅₀ of rMVA-CD40L (Day 0). Mice received 200 μg of IgG2a antibody i.p. at days -2,2,6 and 10; (A) Tumor size follow-up (n= 5 mice/group) and (B) overall survival (n=10 mice/group). (C, D) Performed along with Figures 6C and 6D. B16.OVA tumor-bearing wild type and *IL15ra*^{-/-} mice were grouped according to tumor size. Tumor bearing littermates either received PBS or were immunized with 5x10⁷ TCID₅₀ of rMVA-CD40L (Day 0). Mice received 200 μg of IgG2a antibody i.p. at days -2,2,6 and 10; (C) Tumor size follow-up (n= 5 mice/group) and (D) overall survival (n=10 mice/group). B and D represent overall survival of two merged independent experiments. Data in A and C are expressed as Mean ± SEM. One-way ANOVA was performed on figures A and C. Log rank test on mouse survival was performed for figures B and D. *, P < 0.05; **, P < 0.01; n.s. non-significant.

Second, in a new Figure 6, listed here as Figure 7 Point-by-Point Reply, we show our results in WT, *FcγR*^{-/-} or *IL15ra*^{-/-} B16.OVA tumor bearing mice combining anti-TRP-1 with intravenous rMVA-CD40L. Here, statistical differences in survival among genotypes upon combination of TAA-specific mAb and systemic rMVA-CD40L are easier to depict.

Figure 7 Point-by-point Reply Revision of Figure 6 in the paper, showing that the therapeutic effect of combining tumor-targeting antibodies and rMVA-CD40L is dependent on Fc gamma receptors and relies on NK cells. (A, B) B16.OVA tumor-bearing wild type and *Fc γ R*^{-/-} mice were grouped according to tumor size. Tumor-bearing littermates either received PBS or were immunized with 5x10⁷ TCID₅₀ of rMVA-CD40L (Day 0). Mice received 200 μ g of anti-TRP-1 antibody i.p. at days -2,2,6 and 10; (A) Tumor size follow-up (n= 5 mice/group) and (B) overall survival (n=10 mice/group). (C, D) B16.OVA tumor-bearing wild type and *Il15ra*^{-/-} mice were grouped according to tumor size. Tumor bearing mice either received PBS or were immunized with 5x10⁷ TCID₅₀ of rMVA-CD40L (Day 0). Mice received 200 μ g of anti-TRP-1 antibody i.p. at days -2,2,6 and 10; (C) Tumor size follow-up (n= 5 mice/group) and (D) overall survival (n=10 mice/group). B and D represent overall survival of two merged independent experiments. Data in A and C are expressed as Mean \pm SEM. One-way ANOVA was performed on figures A and C. Log rank test on mouse survival was performed for figures B and D. *, $P < 0.05$; **, $P < 0.01$.

Hence, splitting the data presented in Figures 5E and 5F as suggested allows for a clearer understanding of the involvement of cells expressing Fc gamma receptors in our combination treatment in terms of extended tumor-free survival. In addition, the data obtained using mice devoid of NK cells strongly suggest that this cell population plays a key role in the improvement of therapeutic outcomes in preclinical tumor models achieved by combining TAA mAbs and intravenous administration of rMVA encoding for CD40L. Along with the revised figures from the first manuscript submission, and by addressing the reviewer's suggestions regarding the role of NK cells and B cells in our systems, we believe that we have addressed in detail the mechanism of action of the cellular and molecular components of our systemic rMVA-CD40L and TAA-specific mAb combination setting. From our results we cannot conclude that a specific immune cell subset expressing a Fc γ receptor is involved in the increased efficacy when adding anti-TRP-1 to our vaccination strategy. Since our aim is to employ the novel therapeutic platform described here in clinical trials, we envision a cross-talk between rMVA-CD40-activated Fc γ receptor proficient cells, namely NK cells and phagocytes, and TAA specific mAbs, thereby enhancing objective responses in patients as our proof-of-concept manuscript describes.

The following sentences have been added to the *Discussion* section: "In addition, in the absence of NK cells, the improvement of therapeutic efficacy of combining anti-TRP-1 mAb and systemic rMVA-CD40L immunization was impaired. Therefore, our data reflect the importance of both molecular and

cellular components, namely Fcγ receptors and NK cells respectively, in the therapeutic combination of anti-TAA mAbs and our vaccine platform”.

Minor points:

- The authors should explain clearly at the beginning of Results that they are using MVA expressing OVA or HER (for TAA is explained well). Although this is defined in M&M, the fact that they abbreviate the names of the vectors to only MVA or MVA-CD40L, can create some confusion.

We have addressed this minor comment. The following sentence was introduced at the beginning of Results: “Immunization with a MVA vector encoding ovalbumin (OVA; referred to as rMVA) significantly induced tumor growth control in OVA -expressing B16 melanoma (Figure 1B Manuscript) and EG7.OVA lymphoma (Figure S1A Manuscript) compared to PBS-treated mice. Interestingly, administration of MVA-OVA-CD40L (referred to as rMVA-CD40L) resulted in prolonged mouse survival in melanoma”. We had some inconsistencies, and we are thankful to the Reviewer for picking that up.

- Fig 5C. It is indicated that control groups were treated with anti IgG1 while in Fig. 5D it says IgG2a. According to M&M the anti HER mAb is a human IgG1 mAb. Please correct, because both figures correspond to the same experiment. Please, also include some comment and reference showing that human IgG1 can engage mouse Fc receptors, inducing ADCC.

We would like to thank the reviewer for addressing this point. The figure has been corrected accordingly, and the following sentence has been added to the manuscript: “ Trastuzumab®, the first Food and Drug Administration (FDA)-approved targeted therapy for breast cancer, is a humanized IgG1 monoclonal antibody that targets the HER2 receptor, known to induce ADCC in cells with high HER2 expression⁷, among other mechanism of action. Since human IgG binds mouse FcγR with similar affinity to that of mouse IgG^{8,9}, and human IgG1 induces ADCC with mouse NK cells⁸, we combined anti HER2 treatment with MVA-HER2-CD40L in large CT26.HER2 tumors of volumes above 85 to 100 mm³”.

- Figs. 5E & F are too busy to distinguish well between the different groups. Maybe graphs could be splitted in two, showing in parallel graphs with isotype control IgG and specific anti-TAA mAb, respectively.

As indicated before, graphs have been split in two: on the one hand, the combination of vaccination and isotype control IgG addressing different genotypes in the B16.OVA model has been placed as Supplementary Figure S7 in the manuscript. On the other hand, the combination of TAA-specific mAb anti-TRP-1 and systemic rMVA-CD40L has been placed as Figure 6.

- Fig S4A. Indicate in the axis that you show NK-Ki67 positive cells

Due to restructuring of Figures and Supplementary Figures, Fig S4A is now Fig S5A in the manuscript. In addition, the following changes have been made:

Figure 8 Point-by-point Reply Revision of Supplementary Figure S4A in the paper.

- Some references are not complete (i.e. 5 & 6)

References 5 and 6 have been completed as follows:

Ref 5: Durvalumab Plus CV301 With Maintenance Chemotherapy in Metastatic Colorectal or Pancreatic Adenocarcinoma. www.clinicaltrials.gov NCT03376659 ¹⁰.

Ref 6: A Trial of CV301 in Combination With Anti-PD-1 Therapy Versus Anti-PD-1 Therapy in Subjects With Non-Small Cell Lung Cancer. www.clinicaltrials.gov NCT02840994 ¹¹.

During the revision of this manuscript, the latest results of the Phase I dose escalation trial in NCT02840994 have been published, so the reference has been added to the manuscript ¹².

Reviewer #2 (Remarks to the Author):

The authors investigate the immunostimulatory activity of a modified vaccinia virus Ankara (rMVA) vector containing a CD40L expression cassette in the setting of tumor immunity. The underlying concept is that the overexpression of CD40L may interact with CD40 expressed on professional antigen presenting cells, such as dendritic cells and license them for efficient activation of T cell responses. The current study represents a follow up of a previous study published six years ago, demonstrating that the MVA-CD40L system allows boosting cytotoxic T cell responses (Lauterbach et al., Front. Immunol. 2013). In the current study the authors use the B16.Ova, EG7.OVA, CT26, and CT26 expressing HER2 tumor models and infect tumor bearing mice with rMVA alone or with rMVA-CD40L. By using Batf3^{-/-} mice lacking CD8⁺ dendritic cells the authors show that the priming of T cells may be reduced. Expectedly, the immune cells controlling tumor growth are cytotoxic T cells. Moreover, less pro-inflammatory cytokines seem to be present in Batf3^{-/-} mice upon rMVA-CD40L infection. In addition to T cells the authors also investigate the effect of rMVA-CD40L on NK cell activity and suggest that ADCC activity may be enhanced.

Major experimental concerns:

The authors selectively chose very immunogenic tumor models either expressing OVA or HER2. In the only wildtype model they have included a tumor antigen in their MVA vector, which is not comparable to the other experimental settings. The impact of the study could be enhanced if the authors would compare their data to B16 and EG7 alone.

We really appreciate the reviewer's opinion in raising this important point.

Data obtained in a Bavarian Nordic sponsored CRO study performed at Charles River Discovery Services (Morrisville, NC, USA) using the colon carcinoma model MC38.WT in C57BL/6 mice have been added to Figure 1. Here, single, intravenous administration of rMVA-TAA-CD40L to mice bearing established MC38.WT tumors enhanced tumor growth control. Furthermore, it resulted in tumor-free survival in 2 out of 10 treated individuals. In line with our results in the CT26.WT colon carcinoma model in Balb/c mice, intravenous immunization of rMVA-TAA-CD40L is efficacious in inducing therapeutic responses against cancers that do not overexpress foreign antigens. In addition, this study has been performed using our vaccine in a CRO facility, highlighting the reproducibility of our therapeutic approach.

Therefore, we have edited Figure 1 in the manuscript, showing the following tumor models: B16.OVA, MC38.WT, CT26.WT and CT26.HER2. Our data on EG7, an EL4 clone expressing OVA, have been moved to Supplementary Figure S1.

Figure 1. Therapeutic efficacy of rMVA-CD40L in unrelated, large, established tumor models is independent of the choice of antigen. (A) Experimental layout: Briefly, C57BL/6 (B-E) or Balb/c mice (F-I) received either B16.OVA (B, C), MC38.WT (D, E), CT26.WT (F, G) or CT26.HER2 (H-I) cells subcutaneously in the flank. Seven to fourteen days later tumors were above 60 mm³, mice were immunized intravenously either with PBS or with 5x10⁷ TCID₅₀ of the mentioned rMVA viruses. (B, C) B16.OVA; (B) Tumor size follow-up (n=5 mice/group) and (C) overall survival (n= 10 mice/group) of mice injected either with PBS, rMVA-OVA or rMVA-OVA-CD40L; (D, E) MC38.WT tumor bearing mice were grouped eighteen days after tumor cell inoculation, when tumors were above 90 mm³; (D) Tumor size follow-up (n= 10 mice/group) and (E) overall survival (n= 10 mice/group) of mice injected either with PBS, MVA-TAA or MVA-TAA-CD40L until day 60 after treatment; (F, G) CT26.WT; (F) Tumor size follow-up (n= 5 mice/group) and (G) overall survival (n= 5 mice/group) of mice injected either with PBS, MVA-TAA or MVA-TAA-CD40L; (H, I) CT26.HER2; (H) Tumor size follow-up (n= 6 mice/group) and (I) overall survival (n= 6 mice/group) of mice injected either with PBS or MVA-HER2-CD40L. B, D, F and H data are expressed as Mean ± SEM. B is representative of at least two independent experiments. C, represents overall survival of 2 merged independent experiments. The antitumor efficacy of MVA-HER2-CD40L in Figures H and I has been tested in the CT26.HER2 tumor model in at least two independent experiments. One-way ANOVA at day 20 after tumor inoculation was performed on figures B, D, F and H. Log rank test on mouse survival was performed for figures C, E, G and I. *, P < 0.05; **, P < 0.01; ***, P < 0.005.

The following text has been updated in the *Results* section: “Similar results were observed in MC38.WT, CT26.WT and CT26.HER2 tumor-bearing mice after immunization with a CD40L-adjuvanted MVA encoding restricted endogenous TAA as defined in *Materials and Methods* (rMVA-TAA-CD40L) (Figures 1D and 1F respectively) or human HER2 (MVA-HER2-CD40L, Figure 1H) respectively”.

Figure 10 Point-by-point Reply. Revision of Supplementary Figure S1, showing therapeutic efficacy in EG7.OVA lymphoma and peripheral blood CD8 T cells are increased in rMVA-CD40L immunized tumor-bearing mice. Related to *Figure 1*. C57BL/6 mice received EG7.OVA subcutaneously and experiment was performed as described at *Figure 1A*. Tumor growth was measured at regular intervals. **(A)** Tumor size follow-up (n=5 mice/group) and **(B)** overall survival (n= 10 mice/group) of mice injected either with PBS, rMVA-OVA or rMVA-OVA-CD40L; **(C)** Representative dot plots and frequency of peripheral blood CD44⁺ Tet OVA₂₅₇₋₂₆₄⁺ CD8⁺ T cells 7 days after PBS, rMVA or rMVA-CD40L immunization to B16.OVA tumor-bearing mice (n= 5 mice/group). Data are representative of at least 2 independent experiments. **(D)** Representative dot plots and frequency of peripheral blood CD44⁺ Tet OVA₂₅₇₋₂₆₄⁺ CD8⁺ T cells 7 days after PBS, rMVA or rMVA-CD40L immunization of EG7.OVA tumor-bearing mice (n= 5 mice/group). Data are representative of at least two independent experiments. Data expressed as Mean ± SEM. One-way ANOVA comparing MVA immunization vs PBS. *P < 0.05, ***P < 0.005.

Moreover, the following text has been updated in the *Results* section: “Immunization with a MVA vector encoding ovalbumin (OVA; referred to as rMVA) significantly induced tumor growth control in OVA -expressing B16 melanoma (Figure 1B) and EG7.OVA lymphoma (Supplementary Figure S1A) compared to PBS-treated mice. Interestingly, administration of MVA-OVA-CD40L (referred to as rMVA-CD40L) resulted in prolonged mouse survival in melanoma (Figure 1C) and lymphoma, where

30% of the animals rejected their tumors (Figure S1B). In addition, a strong expansion of OVA₂₅₇₋₂₆₄-specific CD8⁺ T cells was observed in peripheral blood of tumor-bearing mice 7 days after immunization with rMVA vectors in both tumor models (Supplementary Figures S1C and S1D)".

In Figure 1 the rMVA control is not included consistently. As rMVA alone shows strong effects in EG7.OVA for example, having this set of data for all other models would help the interpretation of the data. This concern is of relevance for the entire manuscript (see also later).

We would like to thank the reviewer for pointing this out. Concerning our experimental layouts, the rMVA group was included in the experiment groups. Therefore, all the controls have been added to Figure 1 and consistently throughout the manuscript.

In Figure 3C it is quite evident that tumor growth is enhanced in Batf3^{-/-} at baseline. Thus, the most simple and straightforward explanation for the reduced effects of rMVA-CD40L is that the tumor size is too large to allow a complete control via T cells. In fact there seems to be a significant effect, which the authors do not test for statistical significance. The way the data is shown and discussed is misleading.

Further along these lines it has been demonstrated that Batf3 also plays a role in T cells. As long as the authors do not use a DC or T cell specific Batf3^{-/-} mouse line many of the drawn conclusions remain speculative.

We would like to thank the Reviewer for highlighting a controversial point in the manuscript.

We described in our previous publication that rMVA-CD40L activates IL12p70 production on dendritic cells in the steady state *in vitro* and *in vivo*¹³. Since we observed that CD8⁺ T cells play an important role in our rMVA-CD40L-mediated therapeutic effects, we wanted to understand whether IL12p70 production by MVA would enhance antigen-specific CD8 T cell expansion.

Using a recombinant MVA expressing profilin, which activates TLR11 and TLR12 exclusively in mouse CD8 α ⁺ cDCs resulting in large amounts of IL12p70 secretion, we showed not only IL12p70 secretion in sera upon systemic immunization, but also antitumor responses and enhanced antigen-specific CD8 T cell expansion *in vivo* similar to rMVA-CD40L immunization. With this data, we can claim that CD40L adjuvanted MVA induced IL12p70 production *in vivo*, as so does rMVA-Profilin.

We then used the *Batf3*^{-/-} mice to address whether rMVA-CD40L-induced antitumor effects rely on CD8⁺ T cell cross-priming. Our data in the previous submission demonstrated a strong but not strict dependency on CD8 α ⁺ cDCs by both therapeutic rMVA-CD40L and rMVA-Profilin in a cancer setting. We observed mild tumor growth control in *Batf3*-deficient mice upon rMVA-CD40L or rMVA-Profilin immunization, but the effects of our therapies are not completely abrogated as in our CD8⁺ T cell depletion experiment. It has been described that C57BL/6 *Batf3*^{-/-} mice retain a 3–7% population of CD8 α ⁺ cDCs in spleen and a completely normal population of CD8 α ⁺ cDCs in the inguinal lymph nodes, in contrast to Balb/c *Batf3*^{-/-} or 129SvEv *Batf3*^{-/-} mice¹⁴. In addition, Mittal et al. reported that C57BL/6 *Batf3*^{-/-} bone marrow-derived DCs produce IL12p70 upon overnight LPS stimulation¹⁵. In our experiment, serum IL12p70 levels were undetectable in *Batf3* deficient mice immunized with rMVA-CD40L or rMVA-Profilin compared to wild type littermates 6 hours after immunization, and OVA-specific CD8⁺ T cell frequencies were drastically diminished, but not absent, indicating that other mechanisms of antigen presentation by other DC subsets may apply to our system. Last, the

differential tumor growth among wild type and *Batf3*^{-/-} littermates has been previously observed using similar models in other publications (Sanchez-Paulete et al. *Cancer Discovery* 2015, Figures 1C, 2A-2C¹⁶; Enamorado et al. *Nature Communications* 2017, Figures 7D and 7F¹⁷). Altogether, we agree with the reviewer in the drawbacks of using *Batf3*^{-/-} mice in our experimental settings, thus making the interpretation of our data misleading and speculative.

For the sake of focusing the manuscript on the main message, which is how systemic MVA-CD40L induced innate and adaptive immune responses synergize with tumor targeting antibodies against preclinical cancer models, we decided to omit all the data included in the initial submission concerning the use of *Batf3*^{-/-} tumor bearing mice.

Therefore, Figure 3 as well as Supplementary Figure 3 have been modified accordingly.

Figure 11 Point-by-point Reply. Revision of Figure 3, showing the role of CD8⁺ T cells during rMVA-CD40L-induced tumor growth control

(A-B) CD8⁺ T cell depletion in B16.OVA tumor-bearing mice. When B16.OVA tumors were above 50 mm³, mice received PBS or were immunized with 5x10⁷ TCID₅₀ of either rMVA or rMVA-CD40L. Where indicated, mice received at days -2,2,6 and 10 after immunization 200 μ g of either IgG2b or anti-CD8 antibody i.p.; **(A)** Tumor size follow-up (n= 5 mice/group) and **(B)** overall survival. A shows representative growth of PBS and rMVA-CD40L treated groups at least in 2 independent experiments. B represents overall survival of one independent experiment. Data in A expressed as Mean \pm SEM. **(C-F)** C57BL/6 mice bearing 50 mm³ B16.OVA tumors (n= 5 mice/group) received PBS or were immunized with 5x10⁷ TCID₅₀ of either rMVA,

rMVA-CD40L or rMVA-Profilin; **(C)** Quantitative expression of IL12p70 and IFN- γ in sera 6 hours after systemic immunization; **(D)** Percentage of CD44⁺OVA₂₅₇₋₂₆₄-specific CD8⁺ T cells among peripheral blood leukocytes (PBL) 7 days after immunization; **(E)** Tumor size follow-up; **(F)** overall survival. C-E show data representative of two independent experiments. F represents survival of 2 merged independent experiments. Data in A, C, D and E are expressed as Mean \pm SEM. One-way ANOVA was performed on figures A, C, D and E. Log rank test on mouse survival was performed for figures B and F. n.s. non-significant, *, $P < 0.05$; **, $P < 0.01$; ***, $P < 0.005$.

The *Results* section has been modified accordingly:

Antitumor effect of systemic rMVA-CD40L immunization depends on CD8⁺ T cells

We next addressed the contribution of CD8⁺ T cells in the antitumor effect following rMVA-CD40L immunization. Antibody depletion of CD8⁺ T cells resulted in complete loss of tumor growth arrest by either rMVA or rMVA-CD40L immunization (Figures 3A and 3B), pointing to an essential role of CD8⁺ T cells in controlling tumor growth.

Then, we sought to determine how this potent CD8⁺ T cell response is induced. The generation of CD8⁺ T cells against MVA-encoded antigens seems to rely on CD8⁺ T cell cross-priming¹⁸. Activation of CD8 α ⁺ cDCs results in large amounts of IL12p70, which thereby stimulates interferon gamma (IFN- γ) secretion. In addition to rMVA and rMVA-CD40L, rMVA expressing profilin (rMVA-Profilin) -a protein derived from the protozoan parasite *Toxoplasma gondii* (*T. gondii*) that is specifically recognized by mouse CD8 α ⁺ cDCs via TLR11 and TLR12^{19, 20, 21} - was used to immunize tumor-bearing littermates. rMVA-CD40L and rMVA-Profilin immunization resulted in IL12p70 production and increased levels of IFN- γ in mice sera compared to rMVA (Figure 3C). Similar to rMVA-CD40L, significant expansion of OVA₂₅₇₋₂₆₄- specific CD8⁺ T cells in peripheral blood seven days after rMVA-Profilin compared to rMVA was observed (Figure 3D). In addition, systemic immunization of B16.OVA tumor-bearing mice with rMVA-Profilin controlled tumor growth and prolonged mouse survival comparable to that effect of systemic rMVA-CD40L (Figures 3E and 3F).

The following paragraph has been added to the *Discussion* section of the manuscript: “Our results highlight the importance of CD8⁺ T cell induction by rMVA-CD40L immunization in the observed antitumor response. Since MVA has been shown to infect cDCs *in vivo*^{18, 22, 23}, based on our previous publication¹³ we hypothesized that intravenous immunization using rMVA-CD40L could activate, among other immune cell populations, CD8 α ⁺ cDCs that are specialized in antigen cross-presentation in the context of cancer²⁴, induce secretion of IL12p70 and thereby shape potent antitumor immune responses. To test this, we generated a rMVA expressing profilin, a *T. gondii*-derived protein that activates TLR11 and TLR12 exclusively in mouse CD8 α ⁺ cDCs resulting in large amounts of IL12p70 secretion^{20, 25}. Our data show that rMVA-CD40L and rMVA-Profilin promote tumor growth control, increased IL12p70 levels in sera and antigen-specific CD8⁺ T cell expansion to the same extent. To our knowledge, we are the first to report the therapeutic effect of a profilin-adjuvanted viral vector against cancer. Despite its lack of translational relevance, since TLR11 and TLR12 are not functionally expressed in humans, we believe that rMVA-Profilin constitutes a useful tool to depict the specific-contribution of CD8 α ⁺ cDCs for rMVA in mouse models of disease. Since MVA infection modulates multiple innate cell subsets such as cDCs, plasmacytoid DCs (pDCs), macrophages and NK cells^{26, 27, 28, 29}, additional studies would be needed to better address the specific contribution of antigen cross-presenting CD8 α ⁺ cDCs in the context of generation of antitumor immune responses.”

In Figure 3E the authors show differences in cytokine levels. It remains unclear, however, if this is simply due to the MVA infection. Without the proper rMVA control infection the data is hard to interpret.

As previously explained and commented, the missing rMVA control group has been added in all the figures.

In Figure 5 the authors try to make the point that rMVA-CD40L may enhance ADCC via NK cells. However, the authors never show that NK cells contribute to the anti-tumor activity in their specific setting. The literature they cite and some other additional studies show that NK cells are actually not involved in killing B16 tumors in vivo. Moreover, the critical parts of the Figure (5E and F) are hard to read and it seems that no significant differences could be detected.

Along the role of rMVA-CD40L -induced CD8⁺ T cells in our therapeutic setting, we had observed that the strong innate immune response caused by intravenous administration of rMVA-CD40L activated NK cells systemically and within the tumor microenvironment (Figure 4). When systemic rMVA-CD40L immunization was combined with tumor targeting antibodies, namely anti-TRP-1 and anti HER2, tumor growth control was strengthened and mouse survival was enhanced in two unrelated tumor models against B16.OVA and CT26.HER2 established tumors, respectively (Figure 5 from the revised manuscript).

As asked by both Reviewers, Figures 5E and 5F from the previous version of the manuscript, describing the role of FcγR, have been revised, separated and described in pages 4 to 9 from this point-by-point reply. We show that, using Fc gamma receptor tumor-bearing mice, the effect of the vaccine alone induces tumor growth control in *FcγR* as in WT mice (Figure S7A and S7B). When anti-TRP-1 was combined with rMVA-CD40L, the increased tumor growth control observed in WT littermates is lost in *FcγR* tumor bearers (Figures 6A and 6B). We believe that, thanks to the Reviewer's suggestions, the figure became clearer.

There are conflicting reports showing the involvement of specific FcγR in the antitumor effects of anti-TRP-1 as a monotherapy, and we have cited them in the previous version of this manuscript. Whereas Albanesi et al describe a critical role for FcγRIII in experimental lung metastases of B16 melanoma ³⁰, Otten et al. reported a redundant role for FcγRI and FcγRIV in a model of hepatic metastases after intrasplenic injection of B16.F10 tumor cells ³¹. An additional article by Nimmerjahn et al.-not cited in our manuscript- interrogates the role of FcγRIV in B16 subcutaneous tumors ³². These papers have in common that anti-TRP-1 starts the same day that the tumor is implanted, which might have an effect in the seeding of the tumor cells at the site of implantation. Therefore, these results cannot be compared to our observations, since we use established cancers, where for B16.OVA anti-TRP-1 treatment starts not sooner than day 5 after tumor inoculation, and for anti HER2 not sooner than day 15 after tumor inoculation.

In addition, using *IL15ra*^{-/-} tumor-bearing mice we have shown the importance of NK cells as cellular mediators of the TAA-specific mAb and systemic rMVA-CD40L combined treatment. Our data show that, in the absence of IL15 receptor alpha, in conditions where NK cell numbers are drastically reduced, antigen-specific CD8⁺ T cells were expanded upon systemic vaccination with our vaccine platform. Therefore, systemic rMVA-CD40L immunization -induced tumor growth control against B16.OVA tumors seems not to be NK cell dependent. Interestingly, even when antigen-specific CD8⁺ T cells were still expanded upon combined treatment of rMVA-CD40L and anti-TRP-1, the benefits of the combination observed in wild type littermates were lost in the absence of IL15 receptor alpha. This result supports the notion that NK cells are needed for the combination of TAA-specific mAb and our vaccine platform.

Altogether, and thanks to the revision process, our data demonstrate the role for FcγR- expressing cells and for NK cells in our combined treatment of TAA-specific mAb and systemic MVA-CD40L. However, we cannot rule out that other mechanisms come into play. A very recent report shows that FcγR- expressing macrophages are key to the antitumor effect of an anti-TRP-1 based immunotherapy approach, being the role of NK cells needed for the antitumor response, but not their ADCC abilities, tested by using a transgenic mouse line where FcγRIII/IV is floxed from NK cells³³.

Therefore, the following sentence has been added to the *Discussion* section of the manuscript: “In addition, in the absence of NK cells, the improvement of therapeutic efficacy of combining anti-TRP-1 mAb and systemic rMVA-CD40L immunization was impaired. Therefore, our data reflect the importance of both molecular and cellular components, namely Fcγ receptors and NK cells respectively, in the therapeutic combination of anti-TAA mAbs and our vaccine platform. However, from our results we cannot infer that NK cell -induced ADCC is key in our combined treatment, but that vaccine-induced NK cell expansion and activation are needed for the “two punch” therapeutic strategy.”

References for the Point by Point Reply

1. Rosser EC, Mauri C. Regulatory B cells: origin, phenotype, and function. *Immunity* **42**, 607-612 (2015).
2. Dokun AO, Kim S, Smith HR, Kang HS, Chu DT, Yokoyama WM. Specific and nonspecific NK cell activation during virus infection. *Nat Immunol* **2**, 951-956 (2001).
3. Swann JB, *et al.* Type I IFN Contributes to NK Cell Homeostasis, Activation, and Antitumor Function. *The Journal of Immunology* **178**, 7540-7549 (2007).
4. Chaix J, *et al.* Cutting Edge: Priming of NK Cells by IL-18. *The Journal of Immunology* **181**, 1627-1631 (2008).
5. Lodolce JP, *et al.* IL-15 receptor maintains lymphoid homeostasis by supporting lymphocyte homing and proliferation. *Immunity* **9**, 669-676 (1998).
6. Lodolce JP, Burkett PR, Boone DL, Chien M, Ma A. T cell-independent interleukin 15 α signals are required for bystander proliferation. *J Exp Med* **194**, 1187-1194 (2001).
7. Arce Vargas F, *et al.* Fc-Optimized Anti-CD25 Depletes Tumor-Infiltrating Regulatory T Cells and Synergizes with PD-1 Blockade to Eradicate Established Tumors. *Immunity* **46**, 577-586 (2017).
8. Overdijk MB, *et al.* Crosstalk between human IgG isotypes and murine effector cells. *J Immunol* **189**, 3430-3438 (2012).
9. Dekkers G, *et al.* Affinity of human IgG subclasses to mouse Fc gamma receptors. *MAbs* **9**, 767-773 (2017).
10. Durvalumab Plus CV301 With Maintenance Chemotherapy in Metastatic Colorectal or Pancreatic Adenocarcinoma. www.clinicaltrials.gov NCT03376659. (ed[^](eds)).
11. A Trial of CV301 in Combination With Anti-PD-1 Therapy Versus Anti-PD-1 Therapy in Subjects With Non-Small Cell Lung Cancer. www.clinicaltrials.gov NCT02840994. (ed[^](eds)).
12. Gatti-Mays ME, *et al.* A Phase 1 Dose Escalation Trial of BN-CV301, a Recombinant Poxviral Vaccine Targeting MUC1 and CEA with Costimulatory Molecules. *Clin Cancer Res*, (2019).
13. Lauterbach H, *et al.* Genetic Adjuvantation of Recombinant MVA with CD40L Potentiates CD8 T Cell Mediated Immunity. *Front Immunol* **4**, 251 (2013).
14. Tussiwand R, *et al.* Compensatory dendritic cell development mediated by BATF-IRF interactions. *Nature* **490**, 502-507 (2012).

15. Mittal D, *et al.* Interleukin-12 from CD103(+) Batf3-Dependent Dendritic Cells Required for NK-Cell Suppression of Metastasis. *Cancer Immunol Res* **5**, 1098-1108 (2017).
16. Sanchez-Paulete AR, *et al.* Cancer Immunotherapy with Immunomodulatory Anti-CD137 and Anti-PD-1 Monoclonal Antibodies Requires BATF3-Dependent Dendritic Cells. *Cancer Discov* **6**, 71-79 (2016).
17. Enamorado M, *et al.* Enhanced anti-tumour immunity requires the interplay between resident and circulating memory CD8(+) T cells. *Nat Commun* **8**, 16073 (2017).
18. Gasteiger G, Kastenmuller W, Ljapoci R, Sutter G, Drexler I. Cross-priming of cytotoxic T cells dictates antigen requisites for modified vaccinia virus Ankara vector vaccines. *J Virol* **81**, 11925-11936 (2007).
19. Yarovinsky F, *et al.* TLR11 activation of dendritic cells by a protozoan profilin-like protein. *Science* **308**, 1626-1629 (2005).
20. Lauterbach H, *et al.* Mouse CD8alpha+ DCs and human BDCA3+ DCs are major producers of IFN-lambda in response to poly IC. *J Exp Med* **207**, 2703-2717 (2010).
21. Koblansky AA, *et al.* Recognition of profilin by Toll-like receptor 12 is critical for host resistance to *Toxoplasma gondii*. *Immunity* **38**, 119-130 (2013).
22. Liu L, Chavan R, Feinberg MB. Dendritic cells are preferentially targeted among hematolymphocytes by Modified Vaccinia Virus Ankara and play a key role in the induction of virus-specific T cell responses in vivo. *BMC Immunol* **9**, 15 (2008).
23. Bathke B, *et al.* CD70 encoded by modified vaccinia virus Ankara enhances CD8 T-cell-dependent protective immunity in MHC class II-deficient mice. *Immunology*, (2017).
24. Zitvogel L, Kroemer G. CD103+ dendritic cells producing interleukin-12 in anticancer immunosurveillance. *Cancer Cell* **26**, 591-593 (2014).
25. Raetz M, *et al.* Cooperation of TLR12 and TLR11 in the IRF8-dependent IL-12 response to *Toxoplasma gondii* profilin. *J Immunol* **191**, 4818-4827 (2013).
26. Altenburg AF, *et al.* Modified Vaccinia Virus Ankara Preferentially Targets Antigen Presenting Cells In Vitro, Ex Vivo and In Vivo. *Sci Rep* **7**, 8580 (2017).
27. Pascutti MF, Rodriguez AM, Falivene J, Giavedoni L, Drexler I, Gherardi MM. Interplay between modified vaccinia virus Ankara and dendritic cells: phenotypic and functional maturation of bystander dendritic cells. *J Virol* **85**, 5532-5545 (2011).

28. Garcia Z, *et al.* Subcapsular sinus macrophages promote NK cell accumulation and activation in response to lymph-borne viral particles. *Blood* **120**, 4744-4750 (2012).
29. Samuelsson C, *et al.* Survival of lethal poxvirus infection in mice depends on TLR9, and therapeutic vaccination provides protection. *J Clin Invest* **118**, 1776-1784 (2008).
30. Albanesi M, *et al.* Cutting edge: FcγRIII (CD16) and FcγRI (CD64) are responsible for anti-glycoprotein 75 monoclonal antibody TA99 therapy for experimental metastatic B16 melanoma. *J Immunol* **189**, 5513-5517 (2012).
31. Otten MA, *et al.* Experimental Antibody Therapy of Liver Metastases Reveals Functional Redundancy between Fc RI and Fc RIV. *The Journal of Immunology* **181**, 6829-6836 (2008).
32. Nimmerjahn F, *et al.* FcγRIV deletion reveals its central role for IgG2a and IgG2b activity in vivo. *Proc Natl Acad Sci U S A* **107**, 19396-19401 (2010).
33. Benonis H, *et al.* High FcγR Expression on Intratumoral Macrophages Enhances Tumor-Targeting Antibody Therapy. *J Immunol* **201**, 3741-3749 (2018).

REVIEWERS' COMMENTS:

Reviewer #1 (Remarks to the Author):

I would like to congratulate the authors for the big effort performed to clarify the numerous points presented by the reviewers. I think the new results really strengthen the manuscript and provide more robust and interesting mechanistic data.

Just a couple of minor points:

- I think that in the abstract the authors should indicate that the rMVA-CD40L vector that they used is also expressing tumor antigens.
- Results from Fig S5 are only cited in the Discussion... I have no problems with this if it is fine for the editors. But in that case the figure number should be corrected in order to be correlative with the order in which is cited (in this case after Fig S6 and S7).

Reviewer #2 (Remarks to the Author):

The authors have addressed the majority of my concerns. They have not, however, appropriately addressed a very important issue in the manuscript, concerning their claim that NK cells are involved in ADCC. While the literature on the relevance of individual FcRs is indeed not clear-cut, it is clear from all the cited literature that not NK cells but rather neutrophils or macrophages are involved in ADCC. So either the authors completely remove the claim that NK cells are involved (they could suggest that FcR+ cells are involved including neutrophils, monocytes, macrophages and NK cells) or they use the appropriate experiment to show that NK cells are involved. The least they could do is to use the FcRIII KO mouse strain. As FcRIII is the only FcR expressed on NK cells, this would provide some more (still not definitive) evidence that NK cells could play a role.

Reviewers' comments:

Reviewer #1 (Remarks to the Author):

I would like to congratulate the authors for the big effort performed to clarify the numerous points presented by the reviewers. I think the new results really strengthen the manuscript and provide more robust and interesting mechanistic data. Just a couple of minor points:

- I think that in the abstract the authors should indicate that the rMVA-CD40L vector that they used is also expressing tumor antigens.

We included Reviewer#1 suggestion in the abstract as follows:

“Therapeutic treatment with rMVA-CD40L **expressing tumor-associated antigens** results in the control of established tumors. The expansion of tumor-specific cytotoxic CD8⁺ T cells is essential for the therapeutic antitumor effects.”

- Results from Fig S5 are only cited in the Discussion ... I have no problems with this if it is fine for the editors. But in that case the figure number should be corrected in order to be correlative with the order in which is cited (in this case after Fig S6 and S7).

We would like to thank Reviewer#1 for picking this point. We have corrected the Supplementary Figures order. Therefore, the previous Supplementary Figure 5, that referred to B cells and B cell-mediated antibodies in rMVA-CD40L antitumor effect, becomes Supplementary Figure 8. Changes are highlighted throughout the text in yellow.

Reviewer #2 (Remarks to the Author):

The authors have addressed the majority of my concerns. They have not, however, appropriately addressed a very important issue in the manuscript, concerning their claim that NK cells are involved in ADCC. While the literature on the relevance of individual FcRs is indeed not clear-cut, it is clear from all the cited literature that not NK cells but rather neutrophils or macrophages are involved in ADCC. So either the authors completely remove the claim that NK cells are involved (they could suggest that FcR+ cells are involved including neutrophils, monocytes, macrophages and NK cells) or they use the appropriate experiment to show that NK cells are involved. The least they could do is to use the FcRIII KO mouse strain. As FcRIII is the only FcR expressed on NK cells, this would provide some more (still not definitive) evidence that NK cells could play a role.

To address Reviewer #2 concern regarding any possible claim from our manuscript in the involvement of NK cells in ADCC, the following changes have been made:

1. Page 11 of the manuscript: the statement that reads “Antibody-dependent cellular cytotoxicity (ADCC) or phagocytosis (ADCP) are innate immune defense mechanisms largely exploited in the clinic by monoclonal antibodies (mAbs) directed against tumor associated antigens (TAAs) expressed in different cancers such as HER2 or CD20. One of the main features of NK cells is the induction of ADCC by engagement of the Fc portion of the IgG by the NK cell Fc receptor CD16 (FcγRIII)” has been modified to
“Antibody-dependent cellular cytotoxicity (ADCC) or phagocytosis (ADCP) are innate immune defense mechanisms largely exploited in the clinic by monoclonal antibodies (mAbs) directed against tumor associated antigens (TAAs) expressed in different cancers such as HER2 or CD20.”
2. The sentence suggested by Reviewer #2 has been accommodated as follows (see highlight in yellow) at page 12 of the manuscript: “Whereas FcγR-deficient tumor-bearing mice responded to systemic rMVA-CD40L immunization (Figures S6A and S6B), these mice did not benefit from the combination with anti-TRP-1 mAb neither in tumor growth (Figure 6A) nor in survival (Figure 6B). **This result suggests the involvement of FcγR-proficient cells, including neutrophils, monocytes, macrophages and NK cells, in the combination of anti-TRP-1 mAb and systemic rMVA-CD40L immunization.** Further studies were conducted to dissect the role of NK cells in the therapeutic efficacy of rMVA-CD40L.”
3. The following paragraph in the Discussion section:” In addition, in the absence of NK cells, the improvement of therapeutic efficacy of combining anti-TRP-1 mAb and systemic rMVA-CD40L immunization was impaired. Therefore, our data reflect the importance of both molecular and cellular components, namely Fcγ receptors and NK cells respectively, in the therapeutic combination of anti-TAA mAbs and our vaccine platform. However, from our experiments we cannot infer that NK cell -induced ADCC is key in our combination, but they reflect that vaccine-induced NK cell expansion and activation are needed for the “two punch” therapeutic effect induced by combining anti-TAA mAbs and systemic rMVA-CD40L vaccine regimen. Hence, the latter results emphasize how the induction of antiviral innate immunity by systemic viral immunization can be employed to positively impact the responses to therapeutic vaccination” has been modified, and a sentence was added (highlighted in yellow) as follows:
“ In addition, in the absence of NK cells, the improvement of therapeutic efficacy of combining anti-TRP-1 mAb and systemic rMVA-CD40L immunization was impaired. **Even though, our data**

suggest that NK cells are involved in the enhancement of antibody mediated anti-tumor activity, we cannot conclude that NK cells directly perform ADCC in our models. Another possible explanation could be that NK cell dependent activation, e.g. via IFN- γ secretion, of other immune cells such as neutrophils or macrophages leads to enhanced FcR mediated killing of antibody coated tumor cells. Therefore, our data reflect the importance of both molecular and cellular components, namely Fc γ receptors and NK cells respectively, in the therapeutic combination of anti-TAA mAbs and our vaccine platform. The latter results emphasize how the induction of antiviral innate immunity by systemic viral immunization can be employed to positively impact the responses to therapeutic vaccination”.